# UI-Ins: Enhancing GUI Grounding with Multi-Perspective Instruction-as-Reasoning

Liangyu Chen[1,2],[*] Hanzhang Zhou[2], Chenglin Cai[2], Jianan Zhang[2], Panrong Tong[2]
Xu Zhang[2], Chen Liu[2], Quyu Kong[2], Yuqi Liu[3], Wenxuan Wang[1]
Yue Wang[2],[†] Qin Jin[1],[†] Steven HOI[2]

[1]Renmin University of China    [2]Tongyi Lab, Alibaba Group    [3]CUHK
yue.w@alibaba-inc.com, qjin@ruc.edu.cn

## Abstract

GUI grounding, which maps natural-language instructions to actionable UI elements, is a core capability of GUI agents. Prior works largely treats instructions as a static proxy for user intent, overlooking the impact of instruction diversity and quality on grounding performance. Through a careful investigation of existing grounding datasets, we find a 23.3% flaw rate in their instructions and show that inference-time exploitation of instruction diversity yields up to a substantial 76% relative performance improvement. In this paper, we introduce the **Instruction-as-Reasoning paradigm**, treating instructions as dynamic analytical pathways that offer distinct perspectives and enabling the model to select the most effective pathway during reasoning. To achieve this, we propose a two-stage training framework: supervised fine-tuning on synthesized, diverse instructions to instill multi-perspective reasoning, followed by reinforcement learning to optimize pathway selection and composition. Our resulting models, UI-Ins-7B and UI-Ins-32B, achieve state-of-the-art results on five challenging grounding benchmarks and exhibit emergent reasoning, selectively composing and synthesizing novel instruction pathways at inference. In particular, UI-Ins-32B attains the best grounding accuracy, scoring **87.3%** on UI-I2E-Bench, **57.0%** on ScreenSpot-Pro, and **84.9%** on MMBench-GUI L2. Furthermore, our model demonstrates strong agentic potential, achieving a **74.1%** success rate on AndroidWorld using UI-Ins-7B as the executor. Our in-depth analysis reveals additional insights such as how reasoning can be formulated to enhance rather than hinder grounding performance, and how our method mitigates policy collapse in the SFT+RL framework. All code and models are released.

## 1 Introduction

Automated agents for graphical user interfaces (GUIs) are an important frontier in the pursuit of artificial general intelligence (AGI) (Wang et al., 2024). Their effectiveness is dependent on GUI grounding, i.e., the task of mapping a natural language instruction to the corresponding actionable UI element in a screenshot or live interface.

The natural language instruction is central to GUI grounding: it is a primary input alongside the GUI screenshot and translates high-level user intent into low-level, executable actions. Consequently, the clarity and precision of instruction directly impact grounding success. However, the impact of grounding instruction has been largely overlooked in prior works. In this paper, we provide a comprehensive analysis covering instruction diversity, quality, and algorithmic strategies, and establish a concrete basis for more effective GUI grounding.

We focus on instruction diversity and reveal a fundamental mismatch: humans flexibly choose the most effective pathway among multiple instructional perspectives, whereas current models are trained in a narrow, fixed style. For example, to express a single intent such as "close a window",

---

[*]Work was done during internship at Tongyi Lab, Alibaba Group.
[†]Corresponding authors.

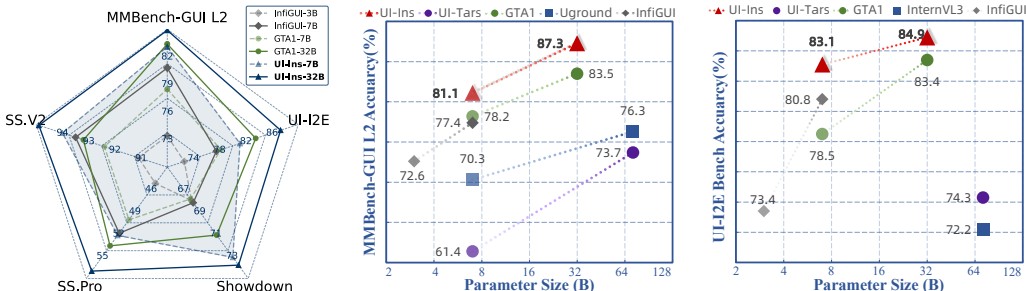

Figure 1: Performance comparisons of UI-Ins and other state-of-the-art methods.

humans may describe the corresponding UI element as its **appearance** ("click the red X"), **function** ("close the file manager"), spatial **location** ("the button in the top-right corner"), or high-level **intent** ("get rid of this screen"). Humans strategically switch among these perspectives, choosing the most effective description for the task at hand, as illustrated in Fig. 3. Our quantitative analysis in Sec. 2.1 likewise shows that leveraging instruction diversity is key to improving grounding accuracy. However, prevailing GUI grounding models are typically trained to map a single instruction style to an action, with limited capacity to reason across different perspectives. This limitation forms a key bottleneck to flexible adaptability and robust interpretation of GUI grounding tasks.

Those insights motivate a paradigm shift: rather than treating instructions as static inputs, we can regard them as *dynamic reasoning pathways*. Different instruction types are not merely alternative phrasings; they encode distinct analytical angles for identifying a UI element. An intelligent GUI agent should not only understand a command but also actively select the most effective reasoning process to infer the user's intent. We term this new paradigm **Instruction-as-Reasoning**.

Beyond this conceptual shift, we also find pervasive instruction quality issues in grounding datasets. Specifically, we manually inspected 1,909 data entries sampled from some popular datasets, including OS-Atlas (Wu et al., 2024a), Widget Captioning (Li et al., 2020), and AMEX (Chai et al., 2025). As shown in Fig. 2b, we found that a notable 23.3% samples contained various quality flaws, introducing considerable noise that could adversely affect model training.

Based on these findings, we introduce a simple yet effective framework. We propose a data pipeline systematically cleans noisy annotations and, crucially, augments existing data with a rich diversity of instruction styles, creating a dataset curated specifically for multi-perspective instruction reasoning. With this high-quality data as our foundation, we then propose our Instruction-as-Reasoning framework. This novel two-stage training paradigm first uses Supervised Fine-Tuning (SFT) to explicitly teach the model to use diverse instruction perspectives as reasoning pathways. Then, it employs Group Relative Policy Optimization (GRPO) (Guo et al., 2025; Shao et al., 2024) in the Reinforcement Learning (RL) stage, enabling the model to learn how to choose the optimal instruction perspective as the reasoning pathway for any given situation. Building on this framework, we introduce the UI-Ins-7B and UI-Ins-32B models. Empirical evaluations conducted across multiple distinct benchmarks validate the strength of our approach, as illustrated in Fig. 1.

To provide additional insights for grounding, we conduct an in-depth analysis of our Instruction-as-Reasoning from multiple perspectives. **First, how can reasoning be formulated to enhance, rather than hinder grounding?** Consistent with prior works (Lu et al., 2025; Tang et al., 2025), we confirm that a free-form reasoning approach often degrades model performance during GRPO. In contrast, experimental results indicate that our proposed Instruction-as-Reasoning consistently enhances performance by a large margin across various base models, establishing it as a highly effective reasoning paradigm for grounding. **Second, how can we mitigate policy collapse in the SFT+RL framework?** We identified that models fine-tuned via SFT using only coordinates as ground truths often exhibit highly uniform responses, leading to ineffective exploration and policy collapse in RL. This is also noted by Phi-Ground (Zhang et al., 2025). However, our Instruction-as-Reasoning framework mitigates this issue by instilling diverse exploratory capabilities after SFT, enabling the model to generate diverse rollouts during RL and thereby avoid policy collapse. **Finally, is UI-Ins's reasoning capability limited to predefined perspectives seen during training?** Interestingly, we observed that after training with Instruction-as-Reasoning, the model not only learns to

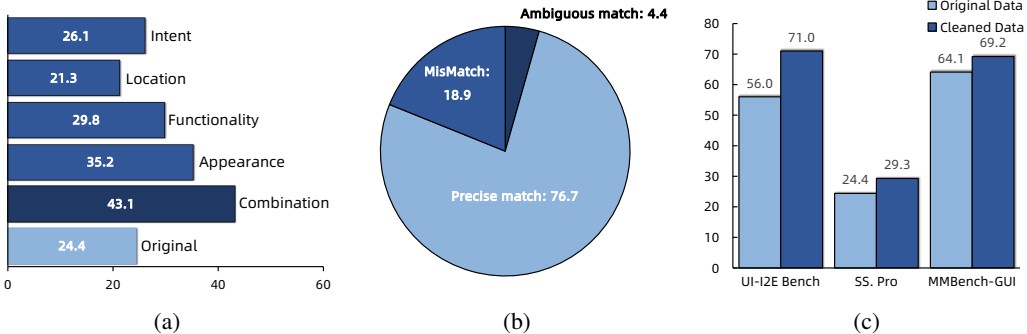

Figure 2: Preliminary analysis of GUI Grounding Instructions. **(a)** Instruction diversity influences performance significantly. **(b)** Instruction quality problems in existing open-source datasets. **(c)** Low instruction quality undermines training efficacy.

select the optimal reasoning pathway but also develops *emergent capabilities* to combine different reasoning perspectives and to reason from novel instruction perspectives not seen in training.

In summary, our contributions are as follows:

- **Systematic Investigation into GUI Grounding Instructions.** We conduct a systematic analysis of instructions in GUI grounding, revealing two crucial insights: (1) a striking **23.3%** of samples' instructions in major datasets are flawed, and (2) there is a massive potential improvement in leveraging instruction diversity, which can unlock up to a substantial **76%** relative performance gain even without training.

- **Instruction-as-Reasoning Paradigm.** Building on insights above, we propose the Instruction-as-Reasoning paradigm, which reframes instructions from static inputs to dynamic reasoning pathways. We realize this through a SFT+GRPO training framework that first teaches the model to use diverse instruction perspectives as reasoning pathways and then incentivizes it to select the optimal analytical reasoning pathway for any given GUI scenario.

- **SOTA Performance Across Diverse Benchmarks.** Our UI-Ins-7B and UI-Ins-32B establish new SOTA performance across **five** most well-known grounding benchmarks. Notably, UI-Ins-32B achieves **87.3%** on UI-I2E-Bench, **57.0%** on ScreenSpot-Pro, and **84.9%** on MMBench-GUI L2, significantly surpassing its strongest counterparts. Moreover, our superior grounding capability leads to strong online agent performance on AndroidWorld when combined with GPT-5 as the planner, yielding a **74.1%** success rate.

- **In-depth Analysis.** Our analysis provides additional insights for grounding. We demonstrate how reasoning can be formulated to augment rather than hinder performance and how our method mitigates policy collapse in the SFT+RL framework. Furthermore, we reveal that our approach unlocks emergent reasoning capabilities, allowing the model to reason from novel perspectives.

## 2 How Much Do Instructions Really Matter?

The natural language instruction is a primary input to grounding tasks, serving as the sole carrier of high-level intent in GUI grounding. But to what extent do the key aspects of an instruction's formulation, namely its analytical perspective and its correctness, truly impact a model's performance? Prior works have largely treated the instruction as a simple input string, leaving its impact underexplored. We highlight that the instruction is a central, understudied variable in GUI grounding. To probe this view, we conduct a preliminary analysis guided by two foundational research questions:

- **RQ1**: How does the **diversity** of instructional perspectives affect grounding accuracy?
- **RQ2**: What is the state of instruction **quality** in GUI grounding datasets, and what is its impact?

### 2.1 Does Instruction Diversity Unlock Higher Performance?

Humans instinctively choose the most effective way to describe an object based on the context like Fig. 3. Does providing a model with similarly diverse, perspective-rich instructions unlock better

performance? To investigate this, we conducted a controlled experiment on the ScreenSpot-Pro benchmark. We systematically rewrote its original instructions to reflect four distinct perspectives: Appearance, Functionality, Location, and Intent. We then evaluated the zero-shot performance of Qwen2.5-VL-7B on each instruction set.

The results, shown in Fig. 2a, reveal two critical insights. First, instruction diversity matters significantly. Instructions from perspectives of appearance, function, and intent all substantially outperform the original instructions. This demonstrates that *even without retraining, simply providing diverse instruction perspectives can unlock significant latent capabilities within the model*. Second, *the ability to select the most appropriate instruction perspective leads to a higher performance ceiling*. The "Combined" bar, representing the performance if a model could always pick the best-performing perspective for each sample, achieves a relative improvement of 76%, far surpassing any single instruction perspective.

Overall, these results reveal considerable untapped potential in leveraging instruction diversity, both by introducing multiple instruction perspectives and by selecting the optimal perspective per instance. This motivates our algorithm that learns to leverage diverse instruction perspectives as reasoning and dynamically chooses the best analytical angle.

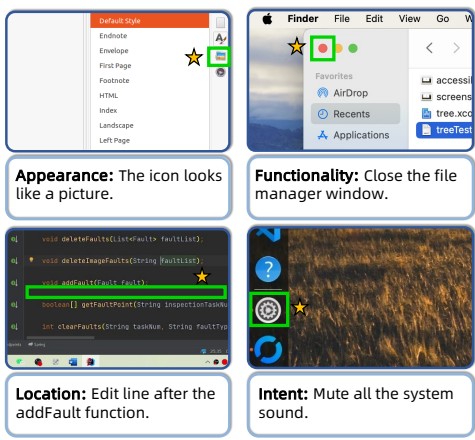

Figure 3: Effective instruction perspectives in different GUI scenarios. Samples are from OS-Atlas Dataset and the ground truth bounding box is labeled with a green box beside a yellow star.

## 2.2 CAN WE TRUST EXISTING DATASETS FOR INSTRUCTION QUALITY?

While utilizing instruction diversity is promising, its effectiveness rests on a foundation that the original instructions are correct. But is this foundation valid? To probe the instruction quality of the grounding datasets, we conducted a large-scale manual analysis. Specifically, we examined 1,909 samples from three prominent datasets, OS-Atlas (Wu et al., 2024a), AMEX (Chai et al., 2025), and Widget Captioning (Li et al., 2020).

Our analysis reveals pervasive instruction quality issues. As shown in Fig. 2b, 23.3% of instructions exhibit substantive flaws, including ambiguity or referring to nothing shown in Fig. 4. To further quantify the impact of such flaws, we trained the same model on the original dataset and on a cleaned version. Experimental results are depicted in Fig. 2c: models trained on cleaned data achieve substantial and consistent performance gains across multiple benchmarks. In other words, flawed instruction data can significantly degrade downstream performance when used for training.

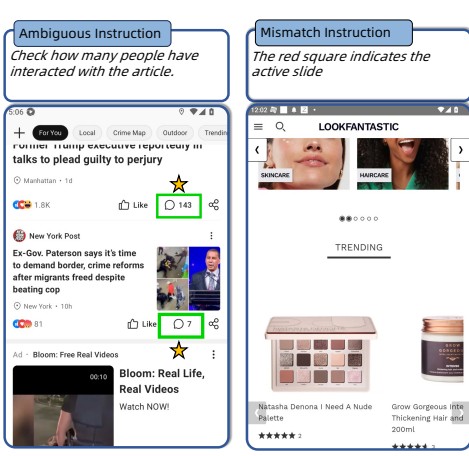

Figure 4: Instruction quality flaws in grounding datasets. **Left:** Ambiguous match, an instruction maps to multi UI elements. **Right:** Mismatch, no valid UI element matches the instruction.

These findings indicate that existing datasets suffer from instruction quality problems that actively harm model performance. Consequently, data cleaning is not optional niceties but necessary prerequisites for meaningful training, especially when our goal is to teach models to leverage diverse instruction perspectives as reasoning.

Figure 5: Overview of our data pipeline. The pipeline first preprocesses the ground-truth box, then leverages GPT-4.1 to generate instructions from diverse perspectives, and finally filter results by ensuring a precise alignment between the instruction and the ground truth box in verification stage.

# 3 METHOD

Our methodology is architected to address the two fundamental challenges identified in Sec. 2: the data quality issues and the untapped potential of instruction diversity. We first introduce a high-quality data pipeline designed to establish the necessary preconditions for effective model training. With this robust data foundation, we then present our core algorithmic contribution, **Instruction-as-Reasoning**, a two-stage training framework that empowers models to use diverse instructions as reasoning pathways and to select the optimal analytical perspective during reasoning.

## 3.1 TASK DEFINITION

GUI Grounding aims to localize a single UI element corresponding to a natural language instruction on a graphical user interface (Wang et al., 2024). Formally, given a screenshot $\mathbf{S}$ and an instruction $\mathbf{I}$, the model $f$ should predict a point $\mathbf{p} = (x_p, y_p)$ that indicates the target element's location.

## 3.2 DATA PIPELINE FOR MULTI-PERSPECTIVE REASONING

Our preliminary analysis (Sec. 2) revealed that data quality is a prerequisite for meaningful training (Sec 2.2) and that instruction diversity unlocks significant performance gains (Sec. 2.1). To this end, we developed a data processing pipeline focused on two primary objectives: establishing a clean data foundation and then systematically augmenting it with diverse, multi-perspective instructions.

**Pre-processing.** To rectify the pervasive annotation noise found in existing datasets, we first perform a lightweight pre-processing step. We use OmniParser V2 (Lu et al., 2024) to detect all UI elements on a screenshot and apply a simple IoU-based method to refine or filter the original ground truth bounding box. This ensures each instruction is associated with a reliable spatial anchor, and the fflawedinstructions are filtered at the same time. The pre-processing forms the clean foundation necessary for the subsequent augmentation.

**Multi-Perspective Instruction Augmentation.** The core of our pipeline focuses on enriching instruction diversity. We leverage GPT-4.1 (OpenAI, 2025a) to generate new instructions from the four fundamental analytical perspectives identified in our analysis: **appearance**, **functionality**, **location**, and **intent**. For each data instance, the model receives the screenshot with the highlighted target element and is prompted to create a set of high-quality, diverse phrasings. To mitigate LLM hallucinations and ensure a strict one-to-one mapping, each generated instruction undergoes a verification step where GPT-4.1 confirms it unambiguously refers only to the target element. This process yields a high-quality, multi-perspective corpus specifically curated to teach complex reasoning.

## 3.3 INSTRUCTION-AS-REASONING

With such a multi-perspective dataset at hand, we introduce the framework to use it. As discussed in Sec. 2.1, leveraging diverse instruction perspectives and dynamically choosing the best analytical angle are key to unlock superior grounding performance. As shown in Fig. 6, our **Instruction-as-Reasoning** framework is a two-stage training approach that instills this capability: (i) a SFT stage that teaches the model to use multi-perspective instructions as explicit reasoning pathways, and (ii) a RL stage that trains the model to use the optimal analytical angle for each sample.

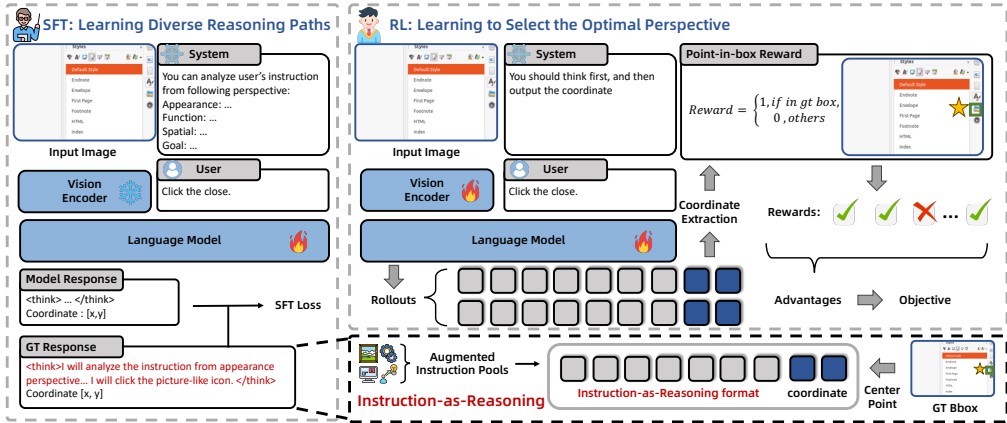

Figure 6: Overview of **Instruction-as-Reasoning**. We leverage diverse instructions as explicit reasoning pathways to teach model multi-perspective reasoning paths in SFT stage, then let model explore unconstrained perspectives to find the optimal ways in different scenarios.

### 3.3.1 SFT Stage: Learning to Generate Diverse Reasoning

The goal of the SFT stage is to explicitly instill the model with the ability to perform **Instruction-as-Reasoning**: utilizing diverse instruction perspectives as analytical reasoning before predicting the grounding coordinate point. Concretely, the model first generates an intermediate reasoning text, i.e., a rewritten instruction from one instruction perspective, which serves as an actionable reasoning pathway (Fig. 6). Then outputs the final coordinate point.

Given the grounding model with parameters $\theta$, the training objective in SFT stage is to maximize the log-likelihood of the target sequence $\mathbf{Y_{gt}}$ across the entire dataset $\mathcal{D}$, formally expressed as:

$$\max_{\theta} \sum_{(\mathbf{S},\mathbf{I},\mathbf{Y}_{gt})\in\mathcal{D}} \log P(\mathbf{Y}_{gt}|\mathbf{S},\mathbf{I};\theta), \quad \text{where } \mathbf{Y}_{gt} = \mathbf{R}_{gt} \oplus \mathbf{p}_{gt} \tag{1}$$

In this formulation, $\oplus$ denotes sequence concatenation. The ground-truth reasoning text, $\mathbf{R}_{gt}$, is randomly sampled from one of the valid augmented instruction perspectives, while $\mathbf{p}_{gt}$ represents the ground-truth coordinate point. An example of SFT prompt and answer is in Sec **??**. This unified objective elegantly compels the model to co-optimize two distinct but related skills:

- **Reasoning Generation:** Learning to produce a reasoning ($\mathbf{R}_{gt}$) in an instruction perspective.
- **Grounded Prediction:** Learning to predict the correct coordinate point ($\mathbf{p}_{gt}$) conditioned on both inputs and its self-generated reasoning.

By fine-tuning on this objective, the model learns to reason from diverse instruction perspectives, building a foundational for RL stage training.

### 3.3.2 RL Stage: Learning to Select the Optimal Perspective

The SFT stage equips the model with the ability to reason from multiple perspectives. However, it does not teach model *which* reasoning pathway is better. To transcend this limitation and incentivize the model to dynamically select the most effective analytical perspective, we introduce an RL stage.

The goal of this stage is to fine-tune the SFT-trained model to discover and select reasoning strategies that maximize grounding accuracy. To achieve this, we employ Group Relative Policy Optimization (GRPO) (Guo et al., 2025). In this phase, we modify the prompt to simply ask the model to "think" before answering, without providing the explicit list of predefined perspectives (appearance, function, etc.). This open-ended instruction encourages the model to explore a wider space of reasoning patterns, including synthesizing multiple perspectives or even formulating entirely novel ones. The model then learns to select the optimal analytical perspective from the feedback of RL rewards.

We calculate rewards by a point-in-box function, then, the rewards $\{r_i\}_{i=1}^{G}$ are normalized by:

$$\hat{A}_{i,t} = \frac{r_i - \frac{1}{G}\sum_{i=1}^{G} r_i}{\sqrt{\frac{1}{G}\sum_{i=1}^{G}\left(r_i - \frac{1}{G}\sum_{i=1}^{G} r_i\right)^2}} \tag{2}$$

Table 1: Overall performance on **MMBench-GUI L2** and **UI-I2E-Bench** benchmarks. We use '-' to denote unavailability, and '*' to denote the results evaluated by us.

| Model | Size | MMBench-GUI L2 | | | UI-I2E-Bench | | |
|---|---|---|---|---|---|---|---|
| | | Basic | Advanced | Avg. | Explicit | Implicit | Avg. |
| Qwen2.5-VL (Bai et al., 2025) | 7B | 38.0 | 29.8 | 33.9 | 58.4 | 51.0 | 53.8 |
| OS-Atlas (Wu et al., 2024a) | 7B | 52.8 | 30.1 | 41.4 | 63.2 | 55.8 | 58.6 |
| SEGUI(todo) (Xu et al., 2025) | 7B | 51.0 | 40.5 | 45.7 | 61.1 | 48.4 | 53.2 |
| Uground-V1 (Gou et al., 2025) | 7B | 78.4 | 53.0 | 65.7 | 81.3 | 63.6 | 70.3 |
| UI-TARS-1.5 (Seed, 2025) | 7B | 78.4 | 50.4 | 64.3 | 81.3 | 68.2 | 73.2 |
| UI-TARS (Qin et al., 2025) | 7B | - | - | - | 71.4 | 55.3 | 61.4 |
| GUI-Actor (Wu et al., 2025) | 7B | 85.7* | 67.5* | 76.5* | 71.6* | 66.1* | 68.2* |
| GUI-G2 (Tang et al., 2025) | 7B | 86.8* | 70.9* | 78.8* | 82.1* | 67.5* | 73.1* |
| InfiGUI-G1 (Liu et al., 2025d) | 7B | 88.5 | 73.2 | 80.8 | 85.0 | 72.7 | 77.4 |
| GTA1 (Yang et al., 2025) | 7B | 84.4* | 72.6* | 78.5* | 87.0* | 72.8* | 78.2* |
| GTA1 (Yang et al., 2025) | 32B | 89.0* | 77.9* | 83.4* | 91.4* | 78.7* | 83.5* |
| Qwen2.5-VL (Bai et al., 2025) | 32B | 80.6* | 63.8* | 72.1* | 73.8* | 62.7* | 66.9* |
| Qwen2.5-VL (Bai et al., 2025) | 72B | 79.7* | 66.7* | 73.2* | 71.3* | 61.2* | 65.0* |
| Uground-V1 (Gou et al., 2025) | 72B | - | - | - | 84.5 | 71.3 | 76.3 |
| UI-TARS (Qin et al., 2025) | 72B | - | - | - | 80.9 | 69.4 | 73.7 |
| **UI-Ins-7B** | 7B | 89.0 | 77.3 | 83.1 | 88.9 | 76.3 | 81.1 |
| **UI-Ins-32B** | 32B | **90.5** | **79.4** | **84.9** | **92.9** | **83.9** | **87.3** |

where $G$ is the rollout number. Finally, the model is optimized by minimizing the objective:

$$L = -\frac{1}{G} \sum_{i=1}^{G} \frac{\pi(o_i \mid I, S)}{\pi_{\text{old}}(o_i \mid I, S)} \cdot \hat{A}_{i,t} \tag{3}$$

where $\pi_{\text{old}}(\cdot \mid \cdot)$ denotes the old policy and $\hat{A}_{i,t}$ is the advantage associated with prediction $o_i$. By iteratively applying this process, the model learns to prioritize reasoning pathways that consistently lead to cothe rrect coordinate point, effectively learning an optimal, context-dependent strategy for instruction perspective selection. Interestingly, we find that the model also learns to *combine multiple perspectives and even formulate entirely novel reasoning perspectives* (Sec. 4.5).

## 4 EXPERIMENT AND RESULTS

### 4.1 EXPERIMENTAL SETTINGS

We source data from several public datasets, including OSAtlas (Wu et al., 2024b), Omniact (Kapoor et al., 2024), Android Control (Li et al., 2024), AMEX (Chai et al., 2025), and AgentNet (Wang et al., 2025b), covering diverse operating systems such as Windows, MacOS, Linux, and Android. All data is subsequently processed through our pipeline to ensure quality. We employ Qwen2.5-VL-7B and Qwen2.5-VL-32B as our backbone architectures. For a comprehensive evaluation, we use multiple grounding benchmarks and online agent benchmarks. More details are in Appendix C.1.

### 4.2 GROUNDING RESULTS

As demonstrated in Tab. 1, UI-Ins-32B achieve SOTA performance on MMBench-GUI L2 and UI-I2E Bench which emphasize complex instruction understanding and outperforms against strong baselines with varying instruction complexity. On MMBench-GUI L2, our model shows progressively larger gains on more difficult tasks. For instance, UI-Ins-7B surpasses Qwen2.5-VL-7B by 134.2% on the 'Basic' subset, and this margin increases to 159.4% on the more challenging 'Advanced' subset. This pattern holds on UI-Ins-32B, where UI-Ins-32B's advantage over Qwen2.5-VL-32B grows from 12.3% ('Basic') to a much larger 24.5% ('Advanced'). A similar trend is observed on the UI-I2E-Bench. When compared to GTA1, UI-Ins-32B's performance gain expands from 1.6% on 'explicit' subset to a more substantial 6.6% on 'implicit' subset. The consistent trend of greater improvement on the 'Advanced' and 'implicit' subsets demonstrates that our approach successfully equips the model with enhanced robustness for difficult scenarios.

Furthermore, to provide a broader validation of our models' capabilities, we conduct extensive evaluations on the ScreenSpot-V2, ScreenSpot-Pro, and Showdown benchmarks. As detailed in Tab. 2,

Table 2: Performance comparison on **ScreenSpot-Pro**, **ScreenSpot-V2**, and **ShowDown**.

| Model | Size | ScreenSpot-Pro | | | ScreenSpot-V2 | | | ShowDown |
|---|---|---|---|---|---|---|---|---|
| | | Text | Icon | Avg. | Text | Icon | Avg. | Avg. |
| Qwen2.5-VL (Bai et al., 2025) | 7B | 38.9 | 7.1 | 26.8 | 94.2 | 81.8 | 88.8 | 43.6* |
| OS-Atlas (Wu et al., 2024a) | 7B | 28.1 | 4.0 | 18.9 | 92.5 | 73.3 | 85.1 | 41.1 |
| UI-TARS (Qin et al., 2025) | 7B | 46.0 | 16.0 | 35.7 | 95.4 | 86.6 | 91.6 | 66.1 |
| UI-TARS-1.5 (Seed, 2025) | 7B | - | - | 42.0 | 92.9 | 83.3 | 89.0 | 67.2 |
| GUI-G$^2$ (Tang et al., 2025) | 7B | 64.9 | 18.4 | 47.5 | 96.1 | 89.7 | 93.3 | 70.4* |
| UGround-v1 (Gou et al., 2025) | 7B | 45.2 | 8.1 | 31.1 | 88.1 | 86.8 | 87.7 | 57.8 |
| InfiGUI-G1 (Liu et al., 2025d) | 7B | 69.1 | 24.5 | 51.9 | 97.4 | 88.4 | 93.5 | 68.2* |
| GTA1 Yang et al. (2025) | 7B | 65.5 | **25.2** | 50.1 | 95.7 | 88.1 | 92.4 | 67.9* |
| GUI-Actor (Wu et al., 2025) | 7B | 58.8* | 20.7* | 44.2* | 96.0 | 87.0 | 92.1 | 64.6* |
| SE-GUI (Yuan et al., 2025) | 7B | 61.8 | 22.8 | 43.2 | - | - | 90.3 | |
| GTA1 Yang et al. (2025) | 32B | 65.6 | 28.1 | 53.6 | 97.1 | 88.3 | 93.2 | 71.1* |
| Qwen2.5-VL Bai et al. (2025) | 32B | 62.6 | 24.3 | 48.0 | 97.1 | 88.3 | 93.2 | 71.1* |
| GUI-Owl Yang et al. (2025) | 32B | 65.6 | 28.1 | 53.6 | 97.1 | 88.3 | 93.2 | 71.1* |
| UGround-v1 Yang et al. (2025) | 72B | 49.0 | 11.1 | 34.5 | 97.1 | 88.3 | 93.2 | 71.1* |
| Qwen2.5-VL Bai et al. (2025) | 72B | 69.4 | 27.3 | 53.3 | 97.1 | 88.3 | 93.2 | 62.3* |
| UI-Tars Yang et al. (2025) | 72B | 65.6 | 28.1 | 53.6 | 97.1 | 88.3 | 93.2 | 71.1* |
| **UI-Ins-7B** | 7B | 70.0 | 23.5 | 52.2 | **98.2** | 88.6 | 94.0 | 73.1 |
| **UI-Ins-32B** | 32B | **73.7** | 30.0 | **57.0** | **98.2** | **90.6** | **94.9** | **73.8** |

UI-Ins-32B again achieves SOTA performance, and UI-Ins-7B consistently outperforms other models of a similar scale. We also provide an error analysis and qualitive analysis in Appendix D.

## 4.3 ONLINE AGENT RESULTS

To rigorously evaluate the stability and reliability of grounding models in realistic settings, we employ UI-Ins-7B as the grounding executor under a GPT-5 (OpenAI, 2025b) planner in AndroidWorld (Rawles et al., 2024) online benchmark, where each action must be grounded and executed in a dynamic environment.

As suggested in Tab. 3, despite our simple architecture without extra knowledge guidance in our designed prompts as shown in Appendix C.2, our framework still achieves a 74.1% task success rate, outperforming strong closed-source models such as Gemini 2.5 Computer Use (DeepMind, 2025) and UI-TARS-2 (Wang et al., 2025a). This result demonstrates that UI-Ins provides precise and stable visual grounding, maintaining semantic alignment and action reliability across diverse app layouts and dynamic interface updates.

Table 3: SOTA Performance on AndroidWorld. Our framework achieves this result by using our model as a grounding executor under a GPT-5 planner, surpassing strong baselines including UI-TARS-2 and the Gemini 2.5 Computer Use.

| Model | S.R. |
|---|---|
| OpenAI CUA-o3 (OpenAI, 2025) | 52.5 |
| Gemini 2.5 Computer Use (DeepMind, 2025) | 69.7 |
| UI-TARS-2 (Zhou et al., 2024) | 73.3 |
| InfiGUIAgent (Liu et al., 2025b) | 9.0 |
| Uground (Gou et al., 2025) | 44.0 |
| Aria-UI (Yang et al., 2024b) | 44.8 |
| AgentS2 (Zhou et al., 2024) | 54.3 |
| JT-GUIAgentV2 (China Mobile, 2025) | 67.2 |
| Qwen2.5-VL-7B (GPT-5 as planner) | 50.0 |
| **UI-Ins-7B (GPT-5 as planner)** | **74.1** |

Additionally, UI-Ins-7B grounding executor achieves a substantial 24.1% performance improvement over its base model (Qwen2.5-VL-7B) in a same configuration. This demonstrates that enhanced grounding capability can effectively translate into improved performance on online agent tasks.

## 4.4 ABLATION STUDY

**Data Pipeline Ablation Study** As shown in Fig. 7a, we manually inspected 1,542 samples generated by our data processing pipeline and found an error rate below 8%. This represents a significant reduction from the 23.3% error rate observed in the original data. To further validate the effectiveness of our data pipeline, we conduct an ablation study using SFT on 210k origin samples, which corresponds to 180k cleaned samples. As shown in Fig. 7b, our data pipeline provides a consistent performance improvement across multiple benchmarks.

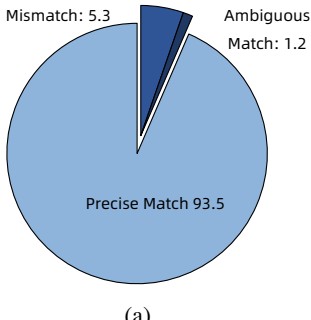
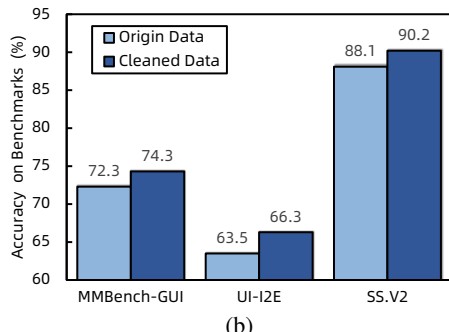

Figure 7: **(a)** Instruction quality distribution after data processing pipeline. **(b)** Performance comparison between Qwen2.5-VL-7B training with origin data and cleaned data by processing pipeline.

**Training Stage Ablation Study** To validate the necessity of SFT+RL training stages for our Instruction-as-Reasoning method. We compare the UI-Ins-7B against two variants: one trained only with SFT and another trained only with RL. In all settings, the model is prompted to generate an intermediate reasoning process. The results of Tab. 4 indicate that both stages are critical for achieving optimal performance. The absence of either stage leads to an accuracy degradation, highlighting the importance of first teaching the model to generate reasoning from diverse perspectives and then allowing it to optimize the selection of the optimal reasoning pathway.

Table 4: Ablation study on training stages. We report accuracy on MMBench-GUI L2 (MM), UI-I2E-Bench (I2E), Showdown (Show), ScreenSpot-Pro (Pro), and ScreenSpot-V2 (V2).

| SFT | RL | MM | I2E | Show | Pro | V2 |
|---|---|---|---|---|---|---|
| ✗ | ✗ | 63.4 | 56.0 | 43.6 | 24.4 | 86.5 |
| ✗ | ✓ | 72.4 | 69.2 | 66.6 | 37.0 | 88.6 |
| ✓ | ✗ | 76.3 | 70.1 | 67.5 | 37.1 | 90.6 |
| ✓ | ✓ | **83.1** | **81.1** | **73.1** | **52.2** | **94.0** |

## 4.5 DEEPER INSIGHTS INTO INSTRUCTION-AS-REASONING

Having established the strong performance of UI-Ins, we now delve deep into the *Instruction-as-Reasoning* to understand its effectiveness. We investigate several central questions below:

**Is an intermediate reasoning step necessary?** A fundamental question is whether generating intermediate reasoning is essential. To answer this, we conducted an ablation study by completely removing the reasoning generation from both the SFT and RL stages, training the model to predict coordinates directly. Experimental results are depicted in Tab. 5. Compared to our method (the 4th row), removing reasoning (the first row) leads to a substantial performance drop across all benchmarks. This result confirms that the intermediate reasoning is crucial to the success of our approach.

Table 5: Ablation on the intermediate reasoning component. Its removal results in a significant performance degradation across all benchmarks. ✓ represents let the model use Instruction as Reasoning in the corresponding stage.

| SFT | RL | MM | I2E | Show | Pro | V2 |
|---|---|---|---|---|---|---|
| ✗ | ✗ | 79.1 | 70.7 | 66.1 | 44.8 | 91.7 |
| ✗ | ✓ | 78.8 | 71.6 | 68.4 | 48.0 | 92.0 |
| ✓ | ✗ | 81.6 | 76.2 | 72.0 | 47.5 | 93.1 |
| ✓ | ✓ | **83.1** | **81.1** | **73.1** | **52.2** | **94.0** |

**Instruction-as-Reasoning (IR) vs. Free-Form Reasoning (FFR).** Given that reasoning is critical, what kind of reasoning is effective? Prior works (Lu et al., 2025; Yang et al., 2025; Zhou et al., 2025; Tang et al., 2025) have shown that FFR is difficult to optimize and can even degrade performance. We test this hypothesis against IR in Tab. 6. As shown in the top section of the table, applying FFR degrades the performance of both UI-Tars-1.5-7B and Qwen2.5-VL-7B, confirming prior findings. For instance, it causes a 6.4% relative drop in SS.Pro for UI-Tars-1.5-7B. In contrast, the bottom section

Table 6: Comparison between FFR and IR in RL. Our Instruction-as-Reasoning is the key to unlocking effective reasoning for GUI grounding.

| Method | Base Model | SS.Pro |
|---|---|---|
| RL (w/o FFR) | UI-Tars-1.5-7B | 50.1 |
| RL (w/ FFR) | UI-Tars-1.5-7B | 46.9↓ (6.4)% |
| RL (w/o FFR) | Qwen2.5-VL-7B | 36.4 |
| RL (w/ FFR) | Qwen2.5-VL-7B | 36.4↓ (0)% |
| RL (w/o IR) | UI-Tars-1.5-7B | 48.7 |
| RL (w/ IR) | UI-Tars-1.5-7B | 51.2↑ (5.1%) |
| RL (w/o IR) | Qwen2.5-VL-7B | 47.5 |
| RL (w/ IR) | Qwen2.5-VL-7B | 52.2↑ (9.9%) |

shows that training models with IR yields significant increases in accuracy. We can thus conclude from the experiments that unstructured FFR fails to improve, whereas IR is the key to unlock effective reasoning for GUI grounding.

**The Hidden Benefit: Stabilizing SFT+RL.** A critical challenge in SFT+RL training for grounding is the policy collapse issue during RL. We compare our SFT+RL framework with a standard one in this ablation. The standard SFT training provides a poor policy initialization, often causing the model's performance to degrade during RL, as evidenced in the upper part of Tab. 7. In contrast, our instruction-as-reasoning-based SFT acts as a powerful exploratory warm-up. By pre-training the model to generate diverse reasoning pathways, we empower it with a strong exploratory capability, achieving a significant performance increase during RL. This demonstrates that our SFT stage not only teaches the reasoning format, but also enables effective and stable policy optimization in the RL phase.

Table 7: Instruction-as-Reasoning prevents policy collapse in RL and achieves significant accuracy gain in RL. This table contrasts our method with a standard SFT+RL pipeline. Scores after 100 RL steps are reported.

| Method | Base Model | SS.Pro |
|---|---|---|
| SFT (w/o IR) | Qwen2.5-VL-7B | 37.0 |
| + RL | Qwen2.5-VL-7B | 34.9↓ (5.7%) |
| Zero-Shot | JEDI-7B | 39.5 |
| + RL | JEDI-7B | 34.5↓ (12.7%) |
| SFT (w/ IR) | Qwen2.5-VL-7B | 37.1 |
| + RL | Qwen2.5-VL-7B | 46.0↑ (24.0%) |

**Emergent Capabilities: Reasoning Beyond Predefined Perspectives.** Does our framework merely teach the model to use the four predefined perspectives? A qualitative analysis of model responses on the UI-I2E Bench (Appendix D.2) reveals that it learns to reason at a much deeper level. We summarize three key emergent capabilities as following:

- *Strategic Selection:* The model learns to strategically select different reasoning perspectives for different scenarios after RL. As shown in Fig. 9b and top section in Fig. 8, diverse and accurate instruction perspectives are selected.

- *Compositional Integration:* The model often combines multiple perspectives into a single, cohesive reasoning, as shown in middle section in Fig. 8. All 1477 samples of UI-I2E Bench contain 5245 reasoning ways in total, as shown in Fig. 9a. This synthesis is not explicitly taught but emerges as an effective reasoning strategy during RL.

- *Emergent Perspective:* Most impressively, as shown in Fig. 9b, the model is capable of generating entirely new analytical angles beyond the four trained perspectives, such as reasoning from the perspective of group affiliation or UI element state, as demonstrated in Fig. 8.

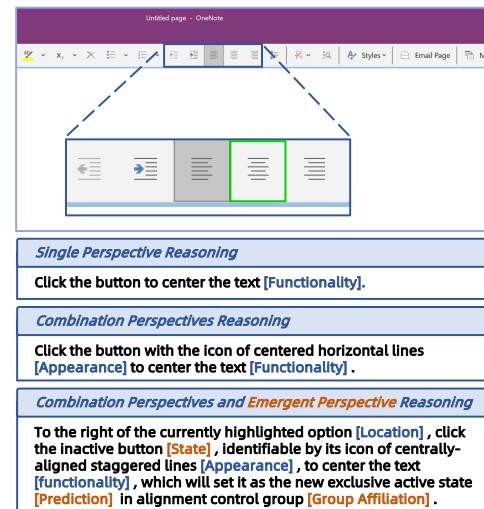

Figure 8: Different reasoning capabilities of UI-Ins.

## 5 CONCLUSION

In this work, we investigate GUI grounding instructions, revealing a 23.3% dataset flaw rate and finding that exploiting instruction diversity yields up to a 76% performance boost. We thus propose **Instruction as Reasoning**, a novel SFT and RL framework that treats diverse perspectives as distinct reasoning pathways. Our models, UI-Ins-7B and UI-Ins-32B, set new state-of-the-art performance across five grounding benchmarks. Specifically, UI-Ins-32B scores **87.3%** on UI-I2E Bench, **57.0%** on ScreenSpot-Pro, and **84.9%** on MMBench-GUI L2. Furthermore, UI-Ins-7B demonstrates strong agentic potential, achieving a **74.1%** execution success rate on AndroidWorld. Our in-depth analysis further reveals helpful insights for GUI grounding.

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

# A  RELATED WORK

## A.1  REASONING IN GUI GROUNDING

Reasoning is a critical capability for MLLMs. However, for GUI grounding task, enabling the model to perform Free-Form reasoning (FFR) during the RL stage does not improve performance and may even degrade it, as demonstrated by GUI-G1 (Zhou et al., 2025), GTA1 (Yang et al., 2025), GUI-G2 (Tang et al., 2025), UI-R1 (Lu et al., 2025), and our own experiments in Sec. 4.5. Although some works, such as InfiGUI-G1 (Liu et al., 2025d), InfiGUI-R1 (Liu et al., 2025c), and GUI-R1 (Luo et al., 2025), have utilized the Free-Form Reasoning, they did not provide ablation experiments to investigate its effectiveness. Furthermore, GUI-R1 found that the model performance is improved as the reward weight for the "thinking" format was decreased. Nevertheless, the failure of free-form reasoning does not imply that reasoning is ineffective for GUI grounding. Our **Instruction-as-Reasoning** method instills strong exploratory capabilities by training the model with diverse and effective reasoning pathways during the SFT stage. Consequently, the model generates more diverse rollouts during RL, effectively mitigating policy collapse.

## A.2  INSTRUCTION IN GUI GROUNDING

Comprehending the user instruction is critical for achieving success in GUI grounding. Prior works, such as Aria-UI (Yang et al., 2024b) and Phi-Ground (Zhang et al., 2025), have primarily focused on augmenting instructions at the input level, using advanced MLLMs to paraphrase them into varying styles. Yet, this methodology suffers from critical limitations: (1) it treats instructions merely as static inputs rather than dynamic reasoning pathways; (2) it lacks a deep analysis of the impact of instruction on grounding; (3) it fails to demonstrate significant, consistent performance gains. Differing from these approaches, our work provides an in-depth investigation of how the diversity and quality of instructions can affect model performance. Moreover, our **Instruction-as-Reasoning** not only enhances the diversity of instructions but also innovatively repurposes these diverse instructions as learnable reasoning pathways for the model, leading to substantial performance improvements.

## A.3  TRAINING PARADIGM IN GROUNDING

Prior GUI grounding methods mainly focus on training in a Supervised Fine-Tuning (SFT) paradigm, such as JEDI (Xie et al., 2025), OS-Atlas (Wu et al., 2024b), Aguvis (Xu et al., 2025), Uground (Gou et al., 2025) and Aria-UI (Yang et al., 2024b). Reinforcement learning methods, particularly GRPO (Guo et al., 2025) have demonstrated remarkable sucess on various visual-language tasks, including Semantic Segmentation (Liu et al., 2025e), Visual Question-Answering (Liu et al., 2025f; Huang et al., 2025) and Temporal Video Grounding (Wang et al., 2025c). Consequently, recent efforts have increasingly focused on adapting RL for GUI grounding. GUI Grounding methods like GUI-R1 (Luo et al., 2025), GUI-Actor (Wu et al., 2025) and GTA1 (Yang et al., 2025) play as an pioneer role in pure RL paradigm and surpass SFT-based methods by a large margin. However, a key limitation of a pure RL paradigm is that it overlooks the substantial benefit offered by an initial SFT stage. While InfiGUI-R1 (Liu et al., 2025c) achieved success with an SFT+RL framework by reframing GUI grounding as a trajectory-level task that encourages model reflection, the SFT+RL paradigm remains notoriously difficult to implement in practice, which is also demonstrated by Phi-Ground (Zhang et al., 2025) and our experimental findings in Sec. 4.5, SFT+RL framework is prone to policy collapse issues. Our Instruction-as-Reasoning method addresses this gap by leveraging SFT to teach model with broader world knowledge and reasoning format demonstrations, and then utilize the RL stage to further incentivize the model to select the best reasoning pathway, establishing a successful example for the SFT+RL training paradigm.

# B  DATA PIPELINE DETAILS

## B.1  INSTRUCTION DIVERSITY AUGMENTATION

To enhance instructional diversity, we expanded the instruction set based on frequently occurring scenarios, categorizing them into four types: appearance-based, function-based, spatial-based, and intent-based. When leveraging GPT-4.1 to augment instructions from open-source datasets, we

mitigated potential hallucinations arising from poor-quality original instructions. To achieve this, we visually grounded the process by overlaying the ground-truth point or bounding box as a distinct circular or rectangular marker on the input image. We provide our detailed prompt as Prompt 1.

---

**Prompt 1:** Prompt for Instruction Generation

**## Task:**
Generate and Translate Unambiguous Grounding Instructions
**## Input:**
GUI Screenshot: An image of a user interface.
Original Instruction: An initial English instruction.
Highlighted Element: A visual marker e.g., a red <annotation_type> on the screenshot pointing to the target UI element.
— CORE OBJECTIVE —
Your primary task is to first translate the Original Instruction into high-quality Chinese, and then generate four new, distinct types of grounding instructions. For all generated instructions, you must adhere to this critical rule: the instruction must correspond to one and only one element on the entire screen—the one highlighted. Clarity and uniqueness are the top priorities.
— IMPORTANT SAFEGUARD —
The <annotation_type> is a ground-truth annotation provided only for your reference. Your instructions must never refer to the annotation itself.
It is noticeable that the original instruction may can not align with the ground-truth annotation, you should follow the ground-truth annotation first.
**## Instructions Generation Requirements:**
Generate one new, clear, and unambiguous instruction for each of the following four categories.
**Appearance-Based:**
A direct and literal description of the element's visual characteristics (e.g., its text, icon, color, shape). Combine features as needed to ensure the description is completely unique.
**Function-Based:**
A clear description of the element's purpose or the immediate outcome of interacting with it (e.g., "the button used to confirm and save your profile changes").
**Spatial-Based:**
An instruction that identifies the element based on its position relative to other prominent, easily identifiable UI elements (landmarks). The described spatial relationship must lead to a unique location.
**Goal-Based:**
A concise phrase that describes the user's ultimate goal or intent. The user must infer which single UI element on the screen fulfills this goal.
**## Output Format:**
The final output must be a single, well-formed JSON object. The JSON structure should begin with the original instruction and its translation, followed by the newly generated instructions.
Now, please process the following inputs and generate the instructions in the specified JSON format.
**Original Instruction:**
<instruction_here>

---

## B.2 Instruction quality refinement

To verify and filter the quality of both the original and the newly generated diverse instructions, we prompted GPT-4.1 to assess whether each instruction uniquely corresponded to a single element in the GUI screenshot. To mitigate potential model hallucinations during this verification process, we visually grounded the task by overlaying the ground-truth annotation directly onto the input image. We provide our detailed prompt as Prompt 2.

**Prompt 2:** Prompt for Verification

## Task:
Quality Evaluation of a GUI Grounding Datum
## Role:
You are a meticulous Data Quality Analyst specializing in user interface datasets. Your task is to critically evaluate a given data sample for its quality and correctness in a structured, two-step process.
## Input:
GUI Screenshot: An image of a user interface.
Grounding Instruction: An English command intended to guide a user to a specific element.
Ground-Truth Bounding Box: A red box drawn on the screenshot, highlighting the target UI element.
——–-IMPORTANT——–-
Ground-Truth Point: A blue hollow circle drawn on the center of the Ground-Truth Bounding Box, which is the key to help you locate the target UI element, because screenshots usually have other red bboxes which may cause distribution.
## Output Process (Two Steps):
### Step 1: Chain-of-Thought Reasoning
First, you must articulate your reasoning process in plain text. Analyze the input and think step-by-step. Your reasoning should cover the following points:
**Instruction Analysis:**
What specific element does the instruction describe? Identify its key features (text, function, location, etc.), it is important you should locate the target UI element according to the blue hollow circle and the red bbox.
Scan the entire screenshot. Are there any other elements that could match this description, even partially?
Conclude whether the instruction is unique or ambiguous based on this scan.
**Bounding Box Analysis:**
What is the target element identified by the instruction? Does it have the blue hollow circle in the center of the box?
Does the red box tightly enclose this entire target element?
Does the box cut off any part of the element?
Does the box include significant empty space or other unrelated elements?
Conclude whether the bounding box is appropriately sized, too large, or too small.
### Step 2: Final JSON Output
After you have completed your reasoning, provide the final answer as a single, well-formed JSON object. This JSON should be the very last part of your response. Do not add any text after the JSON object.

```
{
    "instruction_evaluation": {
        "reasoning": "<A concise summary of your reasoning from Step 1 about the instruction's uniqueness.>"
        "is_unique": <true_or_false>,
    },
    "bbox_evaluation": {
        "reasoning": "<A concise summary of your reasoning from Step 1 about the bounding box size.>",
        "is_appropriately_sized": <true_or_false>
    }
}
```

Now, please perform this two-step evaluation for the following data.
**Grounding Instruction:**
<instruction_here>

# C  EXPERIMENT DETAILS

## C.1  EXPERIMENTAL SETTINGS

**Data and Implementation Details.** We collect data from several public datasets, including OS-Atlas (Wu et al., 2024a), Omniact (Kapoor et al., 2024), Android Control (Li et al., 2024), AMEX (Chai et al., 2025), and AgentNet (Wang et al., 2025b), covering diverse operating systems such as Windows, MacOS, Linux, and Android. All samples are processed through our pipeline to ensure quality. We employ Qwen2.5-VL-7B and Qwen2.5-VL-32B as our backbone architectures. The training procedure consists of two stages:

- **SFT Stage** We fine-tune the models on approximately 283k instances for one epoch. Each instance is organized as Prompt 3, we show **Instruction-as-Reasoning** with red. To teach the model to reason from diverse instruction perspectives, each training instance is constructed by randomly selecting two distinct instruction perspectives from a set of four (appearance, spatial, function and goal) that we defined. One is designated as the instruction perspective, and the other as the reasoning perspective. We use a global batch size of 256 and a learning rate of 5e-6.

---

**Prompt 3:** SFT Training Data Example

**# System Prompt:** You are a GUI agent. You are given a task and your action history, with screenshots. You need to perform the next action to complete the task.
**## Output Format**
Return a json object with function name and arguments within <tool_call></tool_call> XML tags:

<think>
...
</think>
<tool_call>
{"name": "grounding", "arguments": <args-json-object>}
</tool_call>

<args-json-object> represents the following item of the action space:
**## Action Space**
{"action": "click", "coordinate": [x, y]}

Your task is to accurately locate a UI element based on the user's instructions. The screenshot resolution is height {height} and width {width}.
First, you should carefully examine the screenshot and analyze the user's instructions from one of the following different perspectives:
**Appearance Perspective:** You should prioritize visually scanning the user interface to find an element that precisely matches the label, text color, background, shape, and its relative position to other elements as described in the appearance-based instruction. Your decision-making will be driven by direct pattern matching of these visual attributes.
**Function Perspective:** You should identify which UI element is responsible for achieving the specified system state change or executing the described action, such as switching the media display mode. Your decision-making will be guided by understanding the intended effect or utility of interacting with the element.
**Spatial Perspective:** You should first locate any specified reference points (e.g., another UI element or screen edge) and then precisely pinpoint the target element based on its exact directional and positional relationship within the UI layout. Your decision-making will be based on geometric and relational positioning within the interface.
**Goal Perspective:** You should identify the UI element or sequence of actions that directly leads to the fulfillment of the user's stated ultimate objective. Your decision-making will be driven by finding the most direct and efficient path to achieve the desired end-state for the user.
You should translate the user's instruction into the corresponding instruction, and then provide the final action.

---

> # User Instruction:
> Click on the CSDN bookmark in the bookmarks bar to access the CSDN website.
> # Assistant:
> <think>
> I will analyze this instruction from Appearance-Based perspective, the user's instruction can be represented as : Click the bookmark with the red 'C' icon and the label 'CSDN' in the bookmarks bar.
> </think>
> <tool_call>
> {"name":"grounding","arguments":{"action":"click","coordinate":[588,67]}}
> </tool_call>

- **RL Stage** The GRPO training utilizes 33k instances, expanded to approximately 100k training samples by generating a sample per instruction perspective, each instance is organized as Prompt 4. We leave the analytical perspective unspecified in the prompt to encourage exploration. We adopt a learning rate of 1e-6 and 8 rollouts. The batch size is set to 256 for the 7B model and 128 for the 32B model.

> **Prompt 4:** RL Training Data Example
>
> # System Prompt: You are a GUI agent. You are given a task and your action history, with screenshots. You need to perform the next action to complete the task.
> ## Output Format
> Return a json object with function name and arguments within <tool_call></tool_call> XML tags:
>
> <think>
> ...
> <think>
> <tool_call>
> {"name": "grounding", "arguments": <args-json-object>}
> <tool_call>
>
> <args-json-object> represents the following item of the action space:
> ## Action Space
> {"action": "click", "coordinate": [x, y]}
>
> Your task is to accurately locate a UI element based on the user's instructions. The screenshot resolution is height {height} and width {width}.
> First, you should carefully examine the screenshot and analyze the user's instructions in <think>...<think> tags and then output the coordinate.
> # User Instruction:
> Click on the CSDN bookmark in the bookmarks bar to access the CSDN website.
> # Assistant:
> <think>
> ...
> </think>
> <tool_call>
> ...
> </tool_call>

**Baselines and Metrics.** We compare our method's grounding performance against extensive SOTA baselines. These include models that are primarily trained using supervised fine-tuning, such as Jedi (Xie et al., 2025) and Aguvis (Xu et al., 2025), methods that using RL paradigm, such as GUI-Actor (Wu et al., 2025) and InfiGUI-G1 (Liu et al., 2025d), and influential grounding models such as UI-Tars-1.5-7B (Seed, 2025) and GTA1-7B (Yang et al., 2025).. Besides, we also compare UI-Ins with some agentic frameworks such as AgentS2 (Zhou et al., 2024) and InfiGUIAgent (Liu et al., 2025b) on the online benchmark.

Following prior works (Yang et al., 2025; Liu et al., 2025d; Tang et al., 2025), we evaluate GUI Grounding performance using the point-in-box accuracy. A prediction is considered correct if the predicted coordinate point $p = (x_p, y_p)$ falls within the ground-truth bounding box $b = (x_l, y_l, x_r, y_r)$, where the $(x_l, y_l)$ denotes the top-left corner and $(x_r, y_r)$ represents the bottom-right corner. The accuracy over a test set of size $N$ is formally defined as: Accuracy $= \frac{1}{N} \sum_{i=1}^{N} \mathbb{I}(p_i \in b_i)$ , where $\mathbb{I}(\cdot)$ is the indicator function, which equals 1 if the condition is true and 0 otherwise.

**Evaluation Benchmarks.** We evaluate our method on five widely-used grounding benchmarks and a challenging online agent environment.

- *Grounding Benchmarks:* MMBench-GUI L2 (Xuehui Wang et al., 2025) tests performance on hierarchical instructions, while UI-I2E-Bench (Liu et al., 2025a) focuses on explicit instructions and deeper semantic reasoning for implicit instructions. Showdown (Team, 2025) evaluates instruction-following and low-level control capabilities. ScreenSpot-Pro Li et al. (2025) examines semantic understanding in high-resolution professional softwares. ScreenSpot-V2 (Wu et al., 2024a) is a widely adopted benchmark that evaluates model across different operating systems.

  It is notable that, for a comprehensive evaluation, we show the subset scores of all benchmarks, especially for MMBench-GUI L2 and UI-I2E Bench, which emphasize instruction understanding and difficulty. MMBench-GUI L2 is divided into 'Basic' and 'Advanced' subsets, which are distinguished by their instruction style for the same UI element. 'Basic' instructions provide comprehensive visual features, such as "A rectangular button with a dark purple background," whereas 'Advanced' instructions describe the element's inferred purpose, like "Upgrade your current workspace." Similarly, UI-I2E-Bench contains 'explicit' and 'implicit' subsets, where an explicit instruction might be "Enter your email in the subscription field," while an implicit one requires inference, such as "Click to dispatch the email."

- *Online Agent Benchmark:* To evaluate our model's practical utility in a dynamic setting, we report performance on AndroidWorld (Rawles et al., 2024). Unlike simulated or replayed settings, this benchmark introduces realistic challenges, including UI drift, variable rendering latency, asynchronous state transitions, and stochastic user feedback, which collectively pose a strong challenge to the temporal and spatial consistency of grounding.

## C.2 ONLINE BENCHMARK EVALUATION

For the evaluation of the AndroidWorld benchmark, we develop a simple yet effective agent framework to evaluate the grounding capability of our model in the online environment. Our framework consists of two main agents, a planner (*i.e.*, GPT-5), which serves as the high-level controller to decide the executed action in each step, and an executor (*i.e.*, UI-Ins 7B) that identifies the precise coordinates based on each instruction from the planner. Specifically, during each step, the planner receives the task goal, historical records of thinking and actions in previous steps, and the current screenshot, and produces the next reasoning trace and corresponding structured JSON action that conforms to our pre-defined action space. When the predicted action type is `"click"` or `"long-press"`, the JSON instruction is forwarded to our grounding executor. Then our executor interprets the textual description of the target element (*e.g.*, "blue circle button at top-right") and outputs precise screen coordinates, which then performs the `"click"` or `"long-press"` operation on the Android device. The resulting screen update and execution feedback are sent back to the planner, enabling it to iteratively refine its decisions and complete the task through a perception-action loop. We provide our detailed system prompt as Prompt 5.

---

**Prompt 5:** Prompt for Online Agent Evaluation

**## Task.** Control an Android phone to answer user queries and execute tasks with precise, verifiable actions.

**## Role.** *Android Phone Operator AI*. Responsibilities:
- Retrieve information from the device to answer user questions.
- Perform tasks by executing precise UI actions.

**## Action Framework.** Respond with **EXACT JSON** for one of the following actions:

---

| Action | Description | JSON Example |
|--------|-------------|--------------|
| open_app | Open app from <Available Apps> | `{ "action_type":"open_app", "app_name":"Chrome" }` |
| click | Tap visible element | `{ "action_type":"click", "target":"blue circle button at top-right" }` |
| long_press | Long-press visible element | `{ "action_type":"long_press", "target":"message from John" }` |
| input_text | Type into a field | `{ "action_type":"input_text", "text":"Hello", "target":"message input box" }` |
| answer | Respond to user | `{ "action_type":"answer", "text":"It's 25 degrees today." }` |
| navigate_home | Return to home screen | `{ "action_type":"navigate_home" }` |
| navigate_back | Navigate back | `{ "action_type":"navigate_back" }` |
| scroll | Scroll up/down/left/right | `{ "action_type":"scroll", "direction":"down" }` |
| status | Mark task status | `{ "action_type":"status", "status":"complete" }` |
| wait | Wait for screen update | `{ "action_type":"wait" }` |

## Execution Principles.

1. **Communication Rule**
   - Always use `answer` to reply to users; do not assume on-screen text is sufficient.
   - Follow the user instruction strictly (e.g., single number, True/False, comma-separated items).
   - Do not use `answer` for waiting or loading; use `wait`.

2. **Efficiency First**
   - Choose the simplest valid path.
   - If an action fails twice, try an alternative (e.g., `long_press` instead of `click`).

3. **Smart Navigation**
   - Prefer `open_app` with the available app list over manual navigation.
   - Gather information when needed (e.g., open Calendar to check schedule).
   - For scrolling: direction is inverse to swipe; if scroll fails, try the opposite direction.

4. **Text Operations**
   - Activate the input box before typing.
   - Prefer `input_text` over manual typing.
   - For manipulation: long-press to select, use selection bar (Copy/Paste/Select All), delete by selecting then cutting.

## Current Context.

- User Goal: {goal}
- Previous Actions: {history}
- Available Apps: `["Camera","Chrome","Clock","Contacts","Dialer", "Markor","Tasks","Simple Draw Pro","Simple Gallery Pro","Simple SMS Messenger","Audio Recorder","Pro Expense","Broccoli APP","OSMand","VLC","Joplin","Retro Music","OpenTracks","Simple Calendar Pro","Files","Settings",]`

## Decision Process.

1. Analyze goal, history, and current screen.
2. Determine if the task is complete; output `status` if true.
3. If not complete, choose the most appropriate single action.
4. Output in the exact format below, ensuring the action is valid JSON.

## Output Format.

```
Thought:  I need to open the Chrome app to search for the information...
Action:  { "action_type":"open_app", "app_name":"Chrome" }
```

Table 8: Performance comparison on the **MMBench-GUI L2** benchmark.

| Model | Windows | | MacOS | | Linux | | iOS | | Android | | Web | | Avg. |
|---|---|---|---|---|---|---|---|---|---|---|---|---|---|
| | Bas. | Adv. | Bas. | Adv. | Bas. | Adv. | Bas. | Adv. | Bas. | Adv. | Bas. | Adv. | |
| GPT-4o (OpenAI, 2024) | 1.5 | 1.1 | 8.7 | 4.3 | 1.1 | 1.0 | 5.1 | 3.3 | 2.5 | 1.4 | 3.2 | 2.9 | 2.9 |
| Claude-3.7 (Anthropic, 2024) | 1.5 | 0.7 | 12.5 | 7.5 | 1.1 | 0.0 | 13.7 | 10.6 | 1.4 | 1.4 | 3.2 | 2.3 | 4.7 |
| Qwen-Max-VL (Yang et al., 2024a) | 43.9 | 36.8 | 58.8 | 56.1 | 53.9 | 30.1 | 77.4 | 59.1 | 79.5 | 70.1 | 74.8 | 58.8 | 58.0 |
| ShowUI-2B (Lin et al., 2024) | 9.2 | 4.4 | 24.1 | 10.4 | 25.1 | 11.7 | 29.0 | 19.7 | 17.4 | 8.7 | 22.9 | 12.7 | 16.0 |
| Qwen2.5-VL-7B (Bai et al., 2025) | 31.4 | 16.5 | 31.3 | 22.0 | 21.5 | 12.2 | 66.6 | 55.2 | 35.1 | 35.2 | 40.3 | 32.5 | 33.9 |
| OS-Atlas-7B (Wu et al., 2024a) | 36.9 | 18.8 | 44.4 | 21.7 | 31.4 | 13.3 | 74.8 | 48.8 | 69.6 | 46.8 | 61.3 | 35.4 | 41.4 |
| Aguvis-7B (Xu et al., 2025) | 37.3 | 21.7 | 48.1 | 33.3 | 33.5 | 25.0 | 67.5 | 65.2 | 61.0 | 51.0 | 61.6 | 45.5 | 45.7 |
| UI-TARS-1.5-7B (Seed, 2025) | 68.3 | 39.0 | 69.0 | 44.5 | 64.4 | 37.8 | 88.5 | 69.4 | 90.5 | 69.3 | 81.0 | 56.5 | 64.3 |
| UGround-V1-7B (Gou et al., 2025) | 66.8 | 39.0 | 71.3 | 48.6 | 56.5 | 31.1 | 92.7 | 70.9 | 93.5 | 71.0 | 88.7 | 64.6 | 65.7 |
| GUI-Actor-7B* (Wu et al., 2025) | 80.8 | 55.1 | 81.4 | 60.4 | 64.9 | 41.8 | 94.3 | 82.7 | 93.5 | 79.7 | 89.7 | 72.1 | 76.5 |
| SE-GUI-7B* (Yuan et al., 2025) | 77.5 | 57.7 | 77.1 | 60.7 | 68.6 | 44.9 | 95.5 | 80.0 | 95.5 | 83.7 | 89.7 | 68.8 | 76.6 |
| GTA1-7B* (Yang et al., 2025) | 76.8 | 57.4 | 80.3 | 63.9 | 68.6 | 53.6 | 93.9 | 83.3 | 96.3 | 84.5 | 90.3 | 74.7 | 78.5 |
| GUI-G²-7B* (Tang et al., 2025) | 79.7 | 55.1 | 79.7 | 64.7 | 69.6 | 50.0 | 95.2 | 82.7 | 96.6 | 85.4 | 91.9 | 75.6 | 78.8 |
| InfiGUI-G1-7B (Liu et al., 2025d) | 82.7 | 61.8 | 83.8 | 63.9 | 72.3 | 52.0 | 94.9 | 89.4 | 95.2 | 85.6 | 93.5 | 76.3 | 80.8 |
| Qwen2.5-VL-72B (Bai et al., 2025) | 55.7 | 33.8 | 49.9 | 30.1 | 40.3 | 20.9 | 56.1 | 28.2 | 55.6 | 25.4 | 68.4 | 45.8 | 41.8 |
| Qwen2.5-VL-32B* (Bai et al., 2025) | 73.4 | 49.3 | 76.2 | 57.8 | 61.3 | 33.2 | 91.1 | 80.6 | 90.4 | 80.6 | 81.6 | 65.6 | 72.1 |
| InternVL3-78B (Zhu et al., 2025) | 70.1 | 42.6 | 75.7 | 52.3 | 59.2 | 41.3 | 93.6 | 80.6 | 92.7 | 78.6 | 90.7 | 65.9 | 72.2 |
| UI-TARS-DPO-72B (Qin et al., 2025) | 78.6 | 51.8 | 80.3 | 62.7 | 68.6 | 51.5 | 90.8 | 81.2 | 93.0 | 80.0 | 88.1 | 68.5 | 74.3 |
| GTA1-32B* (Yang et al., 2025) | 82.3 | 66.9 | **89.0** | 74.0 | **73.3** | 52.0 | 96.2 | 88.2 | 95.8 | 88.5 | **95.2** | 79.9 | 83.4 |
| **UI-Ins-7B** | 82.7 | 64.7 | 87.2 | **75.1** | 71.7 | 51.5 | 94.9 | 89.7 | 95.8 | 89.0 | 93.2 | 80.8 | 83.1 |
| **UI-Ins-32B** | **84.9** | **68.4** | 88.4 | 73.4 | 68.6 | **56.1** | 96.5 | 91.2 | 97.2 | 92.4 | 94.8 | 85.1 | **84.9** |

Table 9: Performance comparison on **ScreenSpot-V2** and **ShowDown**.

| Model | ScreenSpot-V2 | | | | | | | ShowDown |
|---|---|---|---|---|---|---|---|---|
| | Mobile | | Desktop | | Web | | Avg. | Avg. |
| | Text | Icon. | Text | Icon. | Text | Icon. | | |
| Phi-ground-7B (Zhang et al., 2025) | 90.2 | 76.4 | 93.6 | 75.9 | 96.5 | 62.0 | 83.8 | 62.5 |
| OS-Atlas-7B (Wu et al., 2024a) | 95.2 | 75.8 | 90.7 | 63.6 | 90.6 | 77.3 | 85.1 | 41.1 |
| UGround-v1-7B (Gou et al., 2025) | 83.6 | 90.5 | 85.8 | 86.3 | 95.5 | 83.2 | 87.7 | 57.8 |
| Qwen2.5-VL-7B (Bai et al., 2025) | 97.6 | 87.2 | 90.2 | 74.2 | 93.2 | 81.3 | 88.8 | 43.6* |
| UI-Tars-1.5-7B (Seed, 2025) | 92.2 | 81.5 | 91.0 | 84.2 | 95.5 | 84.5 | 89.0 | 67.2 |
| SE-GUI-7B (Yuan et al., 2025) | **99.3**\* | 89.1* | 96.4* | 78.6* | 92.7* | 81.3* | 90.8* | 63.6* |
| UI-TARS-7B (Qin et al., 2025) | 96.9 | 89.1 | 95.4 | 85.0 | 93.6 | 85.2 | 91.6 | 66.1 |
| GUI-Actor-7B (Wu et al., 2025) | 97.6 | 88.2 | 96.9 | 85.7 | 93.2 | 86.7 | 92.1 | 64.6* |
| OpenCUA-7B (Wang et al., 2025b) | - | - | - | - | - | - | 92.3 | - |
| GTA1-7B (Yang et al., 2025) | 99.0 | 88.6 | 94.9 | 89.3 | 92.3 | 86.7 | 92.4 | 67.9* |
| GUI-G²-7B (Tang et al., 2025) | 98.3 | **91.9** | 95.4 | 89.3 | 94.0 | 87.7 | 93.3 | 70.4* |
| InfiGUI-G1-7B (Liu et al., 2025d) | 99.0 | **91.9** | 94.3 | 82.1 | **97.9** | 89.2 | 93.5 | 68.2* |
| UI-Venus-7B (Gu et al., 2025) | 99.0 | 90.0 | 96.9 | **90.7** | 96.2 | 88.7 | 94.1 | - |
| Qwen2.5-VL-72B (Bai et al., 2025) | 95.5* | 84.4* | 93.8* | 88.0* | 88.5* | 81.8* | 88.2* | 62.3* |
| Qwen2.5-VL-32B (Bai et al., 2025) | 97.9* | 88.2* | 98.5* | 79.3* | 91.2* | 86.2* | 91.3* | 58.2* |
| GTA1-32B (Yang et al., 2025) | 98.6 | 89.1 | 96.4 | 86.4 | 95.7 | 88.7 | 93.2 | 71.1* |
| OpenCUA-32B (Wang et al., 2025b) | - | - | - | - | - | - | 93.4 | - |
| **UI-Ins-7B** | 99.0 | 90.5 | 97.9 | 81.4 | 97.4 | 91.6 | 94.0 | 73.1 |
| **UI-Ins-32B** | 98.6 | 90.0 | **99.0** | 87.9 | 97.0 | **93.1** | **94.9** | **73.8** |

# D    RESULTS DETAILS

## D.1    GROUNDING BENCHMARK RESULTS

In this section, we provide performance details of all five benchmarks. We report all subset scores for MMBench-GUI L2 (Tab. 8), ScreenSpot-Pro (Tab. 11), UI-I2E Bench (Tab 10), ScreenSpot-V2 (Tab. 9) and ShowDown benchmark (Tab. 9).

Table 10: Performance comparison on the **UI-I2E-Bench** benchmark.

| Model | Grouped by Platform | | | Grouped by Implicitness | | Overall |
| --- | --- | --- | --- | --- | --- | --- |
| | Web | Desktop | Mobile | Explicit | Implicit | |
| OS-Atlas-4B (Wu et al., 2024a) | 54.6 | 19.9 | 58.6 | 51.5 | 39.9 | 44.3 |
| UI-I2E-VLM-4B (Liu et al., 2025a) | 60.9 | 38.9 | 61.4 | 61.9 | 48.3 | 53.4 |
| Uground-V1-2B (Gou et al., 2025) | 66.4 | 49.5 | 59.9 | 72.9 | 47.9 | 57.4 |
| UI-TARS-2B (Qin et al., 2025) | 62.2 | 54.0 | 66.7 | 74.1 | 54.5 | 62.0 |
| Aguvis-7B (Xu et al., 2025) | 45.1 | 47.6 | 60.3 | 61.1 | 48.4 | 53.2 |
| Qwen2.5-VL-7B (Bai et al., 2025) | 56.9 | 41.6 | 61.7 | 58.4 | 51.0 | 53.8 |
| OS-Atlas-7B (Wu et al., 2024a) | 52.2 | 48.9 | 68.1 | 63.2 | 55.8 | 58.6 |
| UI-TARS-7B (Qin et al., 2025) | 56.5 | 58.0 | 65.7 | 71.4 | 55.3 | 61.4 |
| GUI-Actor-7B* (Wu et al., 2025) | 65.2 | 63.2 | 72.9 | 71.6 | 66.1 | 68.2 |
| UI-I2E-VLM-7B (Liu et al., 2025a) | 62.1 | 64.0 | 76.2 | 72.0 | 67.9 | 69.5 |
| Uground-V1-7B (Gou et al., 2025) | 70.8 | 65.7 | 73.5 | 81.3 | 63.6 | 70.3 |
| SE-GUI-7B* (Yuan et al., 2025) | 68.4 | 66.3 | 77.4 | 77.5 | 68.6 | 72.0 |
| GUI-G$^2$-7B* (Tang et al., 2025) | 53.4 | 67.8 | 84.0 | 82.1 | 67.5 | 73.1 |
| UI-TARS-1.5-7B (Seed, 2025) | 79.5 | 68.8 | 74.1 | 81.3 | 68.2 | 73.2 |
| InfiGUI-G1-7B (Liu et al., 2025d) | 84.6 | 66.3 | 83.0 | 85.0 | 72.7 | 77.4 |
| GTA1-7B* (Yang et al., 2025) | 77.5 | 71.3 | 83.5 | 87.0 | 72.8 | 78.2 |
| Qwen2.5-VL-72B (Bai et al., 2025) | 49.0 | 47.2 | 55.3 | 49.6 | 52.5 | 51.4 |
| Qwen2.5-VL-32B (Bai et al., 2025) | 76.7 | 61.2 | 67.5 | 73.8 | 62.7 | 66.9 |
| UI-TARS-72B (Qin et al., 2025) | 77.1 | 69.8 | 75.5 | 80.9 | 69.4 | 73.7 |
| Uground-V1-72B (Gou et al., 2025) | 74.7 | 74.6 | 78.2 | 84.5 | 71.3 | 76.3 |
| GTA1-32B* (Yang et al., 2025) | 93.3 | 77.6 | 84.4 | 91.4 | 78.7 | 83.5 |
| **UI-Ins-7B** | 90.5 | 72.8 | 83.8 | 88.9 | 76.3 | 81.1 |
| **UI-Ins-32B** | **95.7** | **81.9** | **88.2** | **92.9** | **83.9** | **87.3** |

Table 11: Performance comparison on the **ScreenSpot-Pro** benchmark.

| Model | CAD | | Dev. | | Creative | | Scientific | | Office | | OS | | Avg. |
| --- | --- | --- | --- | --- | --- | --- | --- | --- | --- | --- | --- | --- | --- |
| | Text | Icon | Text | Icon | Text | Icon | Text | Icon | Text | Icon | Text | Icon | |
| GPT-4o (OpenAI, 2024) | 2.0 | 0.0 | 1.3 | 0.0 | 1.0 | 0.0 | 2.1 | 0.0 | 1.1 | 0.0 | 0.0 | 0.0 | 0.8 |
| Claude C. (Anthropic, 2024) | 14.5 | 3.7 | 22.0 | 3.9 | 25.9 | 3.4 | 33.9 | 15.8 | 30.1 | 16.3 | 11.0 | 4.5 | 17.1 |
| UI-R1-3B (Lu et al., 2025) | 11.2 | 6.3 | 22.7 | 4.1 | 27.3 | 3.5 | 42.4 | 11.8 | 32.2 | 11.3 | 13.1 | 4.5 | 17.8 |
| ZonUI-3B (Hsieh et al., 2025) | 31.9 | 15.6 | 24.6 | 6.2 | 40.9 | 7.6 | 54.8 | 18.1 | 57.0 | 26.4 | 19.6 | 7.8 | 28.7 |
| Qwen2.5-VL-7B (Bai et al., 2025) | 16.8 | 1.6 | 46.8 | 4.1 | 35.9 | 7.7 | 49.3 | 7.3 | 52.5 | 20.8 | 37.4 | 3.8 | 26.8 |
| GUI-R1-7B (Luo et al., 2025) | 23.9 | 6.3 | 49.4 | 4.8 | 38.9 | 8.4 | 55.6 | 11.8 | 58.7 | 26.4 | 42.1 | 16.9 | 31.0 |
| UI-TARS-7B (Qin et al., 2025) | 20.8 | 9.4 | 58.4 | 12.4 | 50.0 | 9.1 | 63.9 | 31.8 | 63.3 | 20.8 | 30.8 | 16.9 | 35.7 |
| GUI-Actor-7B (Wu et al., 2025) | 47.7 | 9.4 | 59.1 | 15.9 | 59.6 | 16.1 | 70.1 | 25.5 | 69.5 | 41.5 | 55.1 | 19.1 | 44.6 |
| SE-GUI-7B (Yuan et al., 2025) | 51.3 | 14.1 | 68.2 | 19.3 | 57.6 | 9.1 | 75.0 | 28.2 | 78.5 | 43.4 | 49.5 | 25.8 | 47.2 |
| GUI-G$^2$-7B (Tang et al., 2025) | 55.8 | 12.5 | 68.8 | 17.2 | 57.1 | 15.4 | 77.1 | 24.5 | 74.0 | 32.7 | 57.9 | 21.3 | 47.5 |
| OpenCUA-7B (Wang et al., 2025b) | - | - | - | - | - | - | - | - | - | - | - | - | 50.0 |
| GTA1-7B (Yang et al., 2025) | 53.3 | 17.2 | 66.9 | 20.7 | 62.6 | 18.9 | 76.4 | 31.8 | 82.5 | 50.9 | 48.6 | 25.9 | 50.1 |
| UI-Venus-7B (Gu et al., 2025) | 60.4 | 21.9 | 74.7 | 24.1 | 63.1 | 14.7 | 76.4 | 31.8 | 75.7 | 41.5 | 49.5 | 22.5 | 50.8 |
| InfiGUI-G1-7B (Liu et al., 2025d) | 57.4 | 23.4 | 74.7 | 24.1 | 64.6 | 18.2 | 80.6 | 31.8 | 75.7 | 39.6 | 57.0 | 29.2 | 51.9 |
| CogAgent-18B (Hong et al., 2024) | 7.1 | 3.1 | 14.9 | 0.7 | 9.6 | 0.0 | 22.2 | 1.8 | 13.0 | 0.0 | 5.6 | 0.0 | 7.7 |
| UGround-v1-72B (Gou et al., 2025) | 16.8 | 4.7 | 55.8 | 4.8 | 54.0 | 10.5 | 70.8 | 22.7 | 61.0 | 18.9 | 40.2 | 7.9 | 34.5 |
| UI-Tars-72B (Qin et al., 2025) | 18.8 | 12.5 | 63.0 | 17.2 | 57.0 | 15.4 | 64.6 | 20.9 | 63.3 | 26.4 | 42.1 | 15.7 | 38.1 |
| Qwen2.5-VL-32B (Bai et al., 2025) | 34.5 | 20.3 | 74.0 | 22.1 | 61.1 | 16.1 | 75.0 | 30.0 | 74.6 | 30.2 | 64.5 | 33.7 | 50.5 |
| Qwen2.5-VL-72B (Bai et al., 2025) | 54.3 | 14.1 | 78.6 | 26.9 | 62.6 | **20.3** | 77.8 | **34.5** | 80.2 | 47.2 | 67.3 | 28.1 | 53.3 |
| GTA1-32B (Yang et al., 2025) | 43.7 | 23.4 | 82.5 | **28.3** | 69.2 | 14.7 | 79.9 | 31.8 | 80.8 | 43.4 | **70.1** | 32.6 | 53.6 |
| OpenCUA-32B (Wang et al., 2025b) | - | - | - | - | - | - | - | - | - | - | - | - | 55.3 |
| **UI-Ins-7B** | **60.9** | 20.3 | 75.3 | 18.6 | 65.2 | 18.9 | 81.3 | 29.1 | 79.7 | 37.7 | 57.0 | 25.8 | 52.2 |
| **UI-Ins-32B** | 51.8 | 29.7 | **83.1** | 26.9 | **69.7** | 18.9 | **83.3** | **34.5** | **88.7** | **50.9** | **70.1** | **34.8** | **57.0** |

## D.2 REASONING PERSPECTIVE ANALYSIS

We performed a detailed classification of the model's reasoning process by first manually defining ten distinct analytical perspectives. We then utilized GPT-4.1 to examine 1477 responses generated by UI-Ins-7B on the whole UI-I2E benchmark based on the taxonomy as following:

> **Taxonomy:** Reasoning perspectives
>
> **1. Appearance**
> Abbreviation: app
> Definition: Describes the static visual properties of a UI element, including its color, shape, icon, style, and the literal text it displays.
> **2. Functionality**
> Abbreviation: func
> Definition: Describes the element's purpose, its action, or what happens when a user interacts with it.
> **3. Location**
> Abbreviation: loc
> Definition: Describes the element's spatial position on the screen or in the viewport, which can be absolute (e.g., "top-left") or relative to other elements (e.g., "below the title").
> **4. Intent**
> Abbreviation: intent
> Definition: Describes the high-level user goal or plan that motivates the entire action. It is often the starting point of a reasoning chain.
> **5. Structural Relationship**
> Abbreviation: struct
> Definition: Describes the element's position within the UI's layout hierarchy (like a DOM tree), emphasizing its parent, child, or sibling relationship to other elements or containers.
> **6. State**
> Abbreviation: state
> Definition: Describes the current dynamic condition of an element, such as whether it is interactive, active, selected, disabled, or checked.
> **7. Component Type**
> Abbreviation: ctype
> Definition: Identifies the element as a standard, reusable design pattern or component, rather than just describing its appearance.
> **8. Sequential Position**
> Abbreviation: seq
> Definition: Describes the element's order or temporal place within a multi-step user task.
> **9. Salience**
> Abbreviation: salience
> Definition: Describes the element's degree of visual prominence, which is often determined by its size, contrast, unique styling, or animation.
> **10. Accessibility**
> Abbreviation: a11y
> Definition: Describes non-visual properties provided for assistive technologies, such as screen readers. This includes ARIA labels, roles, and other accessibility attributes.

Based on this analysis, we show the statistics details in Fig. 9, as demonstraed in Fig 9a, Instruciton-as-Reasoning help model combines different reasoning perspective during inference, furthermore, ans Fig 9b shows UI-Ins is capable to generate entirely new analytival angles beyond the four trained perspectives.

## D.3 ERROR ANALYSIS

We conducted an error analysis and identified three primary types of failures in the GUI grounding performance of UI-Ins:

- **Lack of Domain-Specific Knowledge:** As shown in Fig. 10 (a), The model's erroneous selection of "Jazwares" demonstrates a failure in real-world knowledge grounding, as it lacks the external knowledge required to associate the abstract description "company known for building block toys" with the correct brand entity, "MEGA".

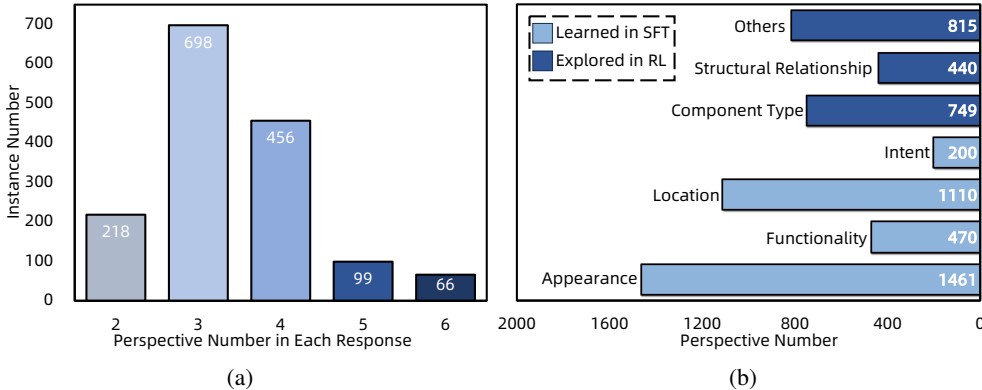

Figure 9: **(a)** UI-Ins combine multiple reasoning pathways in each response. **(b)** UI-Ins can select different reasoning paths and can explore emergent reasoning perspectives after RL.

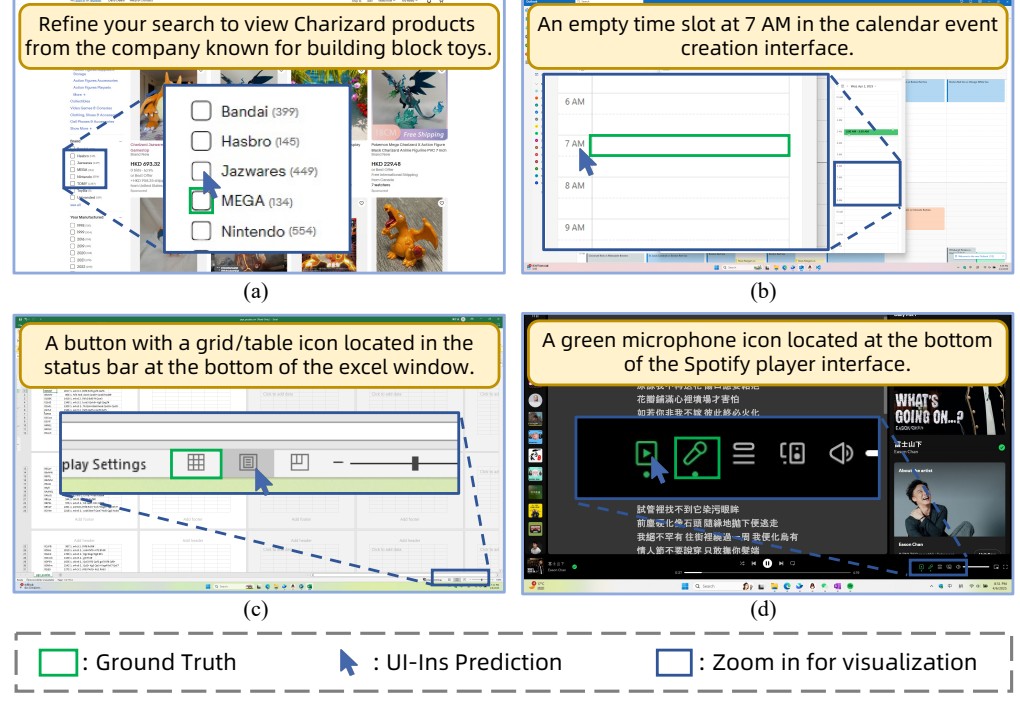

Figure 10: Error analysis of UI-Ins. **(a):** Lack of domain specific knowledge. **(b):** Lack of layout understanding ability. **(c)** and **(d):** Hallucination of MLLMs.

- **Lack of layout understanding ability:** Illustrated in Fig. 10 (b), the model is unable to discern the correct clickable area required to fulfill the instruction, demonstrating a weakness in understanding the structural layout of the user interface.
- **Visual Ambiguity and Hallucination:** As seen in Fig. 10 (c) and (d), when a visually similar distractor icon is present alongside the ground-truth target, the model struggles to disambiguate between them and may select the incorrect one.

## D.4 QUALITATIVE ANALYSIS

To provide a comprehensive qualitative analysis, we compare our UI-Ins-7B and GTA1-7B on ambiguous grounding tasks which need to understand the implicit instructions, as shown in Fig 11, this comparison indicates our Instruction-as-Reasoning approach is key to resolving challenging cases where other models fail. Furthermore, we provide the grounding results of UI-Ins-32B across var-

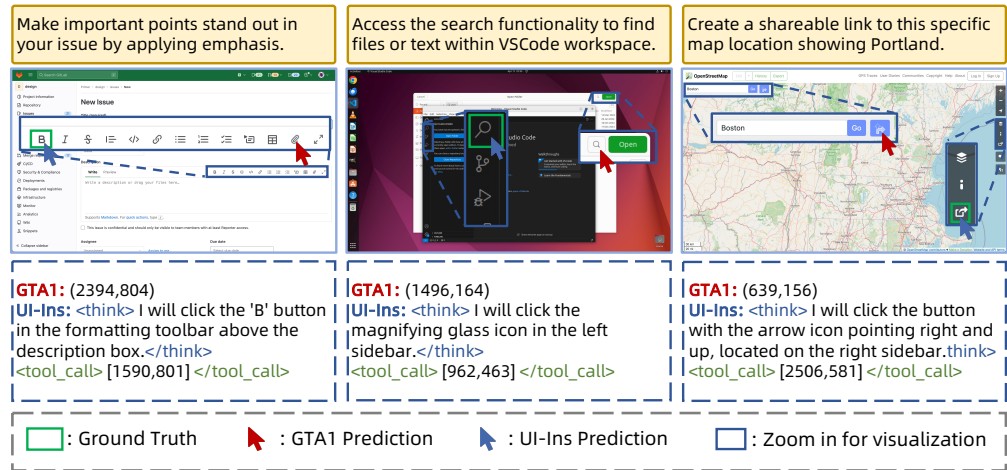

Figure 11: Reasoning from diverse instruction perspectives enables UI-Ins-7B to succeed on ambiguous grounding tasks. This qualitative comparison with GTA1-7B showcases how our Instruction-as-Reasoning process is key to resolving challenging cases where other models fail.

ious platforms and software applications. As shown in Fig. 12, UI-Ins-32B demonstrates robust performance on diverse platforms.

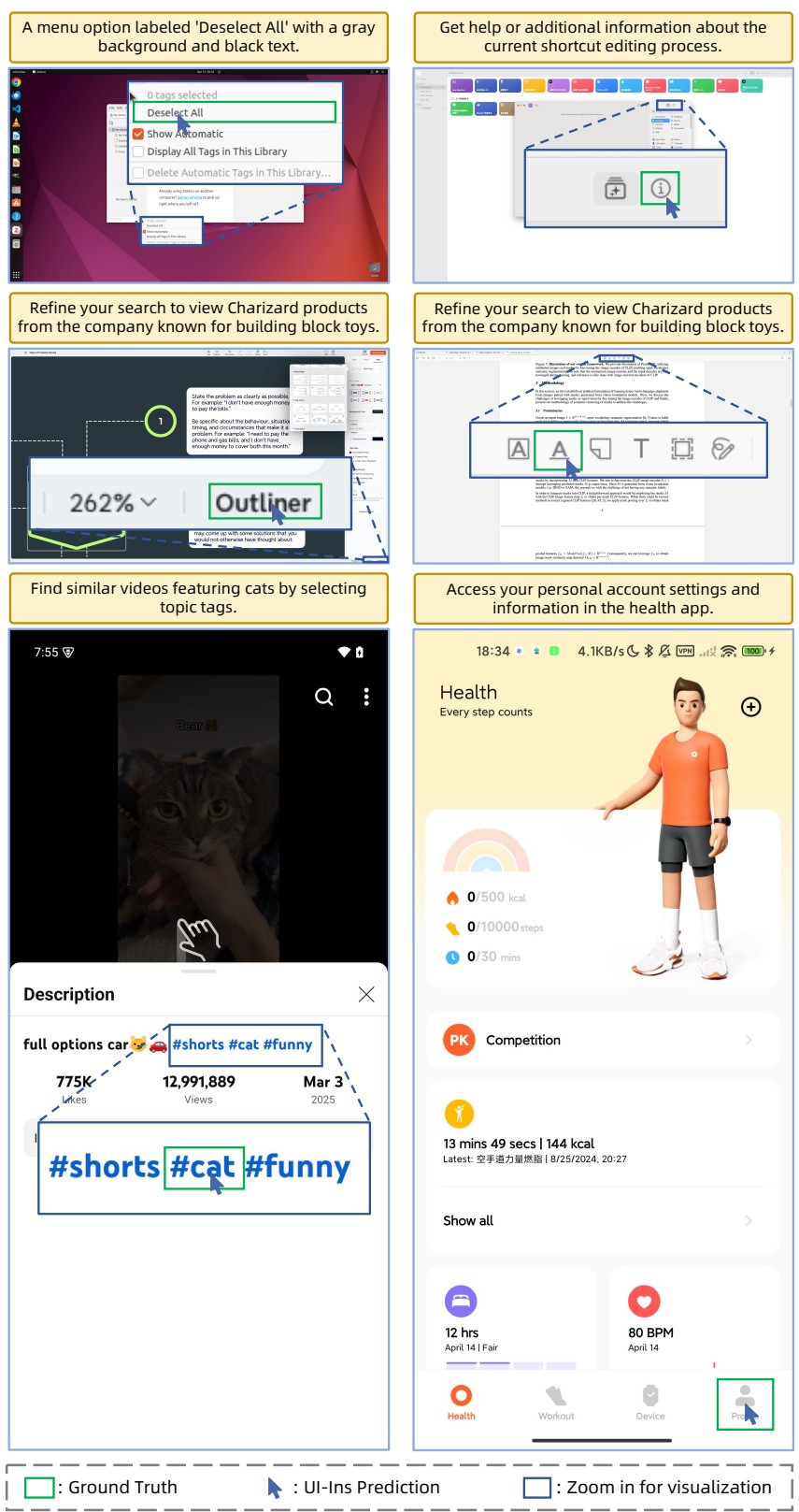

Figure 12: Success Examples of UI-Ins-32B

