# OpenReview forum: "UI-Ins: Enhancing GUI Grounding with Multi-Perspective Instruction as Reasoning"
_ICLR.cc/2026/Conference — ICLR 2026 Poster_

### Official Review · Reviewer_UPmG · 2025-10-26

**Soundness:** 3
**Presentation:** 3
**Contribution:** 2
**Rating:** 6
**Confidence:** 3

**Summary:**

This paper addresses the GUI grounding task by proposing a novel paradigm called "Instruction as Reasoning" (IR). The authors argue that existing methods treat instructions as static user intent, overlooking the value of instruction diversity. The core contribution is a two-stage training framework: first, using Supervised Fine-Tuning (SFT) on a cleaned and augmented dataset (containing multi-perspective instructions) to teach the model to generate intermediate reasoning paths; second, employing a Reinforcement Learning (RL) stage to optimize the model to autonomously select the best reasoning path for a given task. The authors introduce the UI-Ins-7B and UI-Ins-32B models, achieving SOTA performance on five major benchmarks and demonstrating an emergent ability to combine and generate novel reasoning paths in new scenarios. I think this is a well-motivated and experimentally sound paper.

**Strengths:**

1. The models achieve SOTA performance on five benchmarks, including MMBench-GUI L2 and UI-I2E-Bench.
2. The "Instruction as Reasoning" (IR) paradigm is novel. A key insight of the paper is that traditional "Free-Form Reasoning" (FFR) may degrade grounding performance, while the proposed structured IR significantly improves it.
3. The proposed two-stage SFT+RL (GRPO) framework is methodologically sound. The SFT stage effectively teaches the model to understand and generate diverse reasoning paths, while the RL stage optimizes the path selection strategy.
4. The experimental evidence supporting the method is convincing. The ablation studies are strong, validating the necessity of both the SFT and RL stages, the importance of the intermediate reasoning step, and the role of IR-SFT in stabilizing RL training.

**Weaknesses:**

1. **Choice of GRPO**
   The paper employs GRPO for the RL stage but does not sufficiently justify why GRPO was chosen over other reinforcement learning algorithms. A comparative discussion or empirical evidence supporting this choice would strengthen the methodological soundness.

2. **Definition of "Optimal" Selection**
   The paper claims that the RL stage learns to select the "optimal" analysis perspective. However, there is insufficient analysis explaining:
   - Under which contexts the model selects specific perspectives
   - Why these selections are considered "optimal"

   A deeper examination of selection patterns and their correlation with performance would help substantiate this claim.

**Questions:**

1. **Rationale for Selecting GRPO**
   Provide more justification for using GRPO in the RL stage. Include comparisons or discussions with other popular RL algorithms to clarify the advantages or unique suitability of GRPO for this task.

2. **Qualitative Analysis of Emergent Reasoning**
   Expand the qualitative analysis to better illustrate the phenomenon of "emergent reasoning." More detailed case studies are needed to show when and why the model synthesizes multiple perspectives or formulates entirely novel instructional perspectives.

---

> ### Author Response · Authors · 2025-11-24
> **Author response (1/2)**
>
> **Dear Reviewer UPmG:**
>
> We sincerely thank Reviewer UPmG for your valuable feedback and positive assessment. We are encouraged by your recognition of our paper, including **the novelty of the Instruction-as-Reasoning" (IR) paradigm**, **the methodological soundness of the SFT+RL framework**, **The SOTA performance of five benchmarks**, and **the convincing experimental evidence**.
>
> During rebuttal we have carefully responded to each of your valuable feedback, we **summarize our responses below**:
>
> 1. **Insight from RL algorithm analysis: Instruction-as-Reasoning is robust on different RL algorithms**
>
>    Following your suggestion, we compared other RL algorithms for Instruction-as-Reasoning. Experimental results demonstrate IR's robustness across different RL algorithms, and GRPO can achieve better trade-off between performance and efficiency.
>
> 2. **Presentation improvement**
>
>    In light of your valuable feedback, we revised the paper regarding the term “optimal selection” to ensure the rigor of the paper.
>
> In light of your insightful comment, we are encouraged to find that these experiments and analysis have largely strengthen our paper. We thank you for your constructive feedback. And we show details in the following response.
>
> ---
>
>
>
> **[W1 & Q1] Rationale for Selecting GRPO:**
>
> We justify our choice of GRPO through both theoretical constraints and new empirical comparisons with other RL algorithms (RLOO, DAPO).
>
> **1. Theoretical Fit: Why not PPO or DPO?**
>
> *   **vs. PPO:** Standard PPO [1] requires a value model (critic) equal in size to the policy model. For 7B/32B VLMs, this doubles GPU memory usage. GRPO eliminates the Critic, enabling larger batch sizes crucial for stability.
> *   **vs. DPO:** DPO [2] requires pairwise preference data ($y_w, y_l$), which is costly to curate for reasoning paths. GRPO works with rule-based rewards (IoU), fitting our "Instruction-as-Reasoning" goal perfectly.
> *   **vs. Other R1-like methods:** GRPO is a standard setting, we didn't use some training strategies like clip-higher in our training to illustrate that the effectiveness of Instruction-as-Reasoning is not from RL training tricks.
>
> **2. Robustness of Instruction-as-Reasoning in empirical Comparison: GRPO vs. RLOO vs. DAPO.**
> Following your suggestion, we trained UI-Ins-7B using **RLOO** [3] and **DAPO** [4]. The results (first 100 steps) are shown in **Table R1** and **Table R2**.
>
> *   **Performance and Efficiency:** Though DAPO can achieve a higher performance by dynamic sampling (Advantages is higher in theoretical), DAPO is significantly slower (**3x training time**) due to dynamic sampling overhead and exhibited instability (breaking at step 108). RLOO performs similarly to GRPO.
> *   **Conclusion:** We selected GRPO because it offers the **better trade-off between performance and computational efficiency** (913s/step vs 2885s/step for DAPO).
>
> __Table R1. Performance comparison of different RL methods__
>
> | Step         | Method          | MMBench-GUI | UI-I2E   | Showdown | ScreenSpot-Pro | ScreenSpot-V2 |
> | :----------- | :-------------- | :---------- | :------- | :------- | :------------- | :------------ |
> | **Step 25**  | RLOO            | 80.0        | 72.0     | 68.6     | 47.5           | 92.1          |
> |              | DAPO            | **80.7**    | **73.4** | **69.3** | **48.0**       | **92.5**      |
> |              | **GRPO (Ours)** | 79.5        | 72.2     | 68.0     | 47.6           | 92.0          |
> |              |                 |             |          |          |                |               |
> | **Step 50**  | RLOO            | 81.0        | 75.6     | **71.6** | 48.7           | 92.1          |
> |              | DAPO            | **82.3**    | **80.0** | 70.9     | **49.7**       | **93.0**      |
> |              | **GRPO (Ours)** | 81.4        | 77.7     | 70.0     | 47.9           | 92.3          |
> |              |                 |             |          |          |                |               |
> | **Step 75**  | RLOO            | 81.5        | 79.6     | 70.7     | 49.1           | 92.9          |
> |              | DAPO            | **82.7**    | **80.9** | 71.8     | **49.3**       | **93.6**      |
> |              | **GRPO (Ours)** | 82.2        | 79.3     | **72.0** | 48.4           | 92.8          |
> |              |                 |             |          |          |                |               |
> | **Step 100** | RLOO            | 82.6        | 80.1     | 70.0     | 49.7           | **93.2**      |
> |              | DAPO            | **82.8**    | **80.8** | 71.8     | **51.3**       | **93.2**      |
> |              | **GRPO (Ours)** | 82.0        | 78.8     | **72.4** | 50.3           | 92.9          |
>
>
>
> __Table R2. Training effiency of different RL methods__
>
> | Method | avg training time/Step (s) |
> | ------ | -------------------------- |
> | RLOO   | 923.56                     |
> | DAPO   | 2885.47                    |
> | GRPO   | 913.43                     |

---

> ### Author Response · Authors · 2025-11-24
> **Author response (2/2)**
>
> **[W2] Definition of 'Optimal' Selection**
>
> Thank you for your rigorous review of the paper. We agree with the reviewer that "optimal" implies a theoretical guarantee that is difficult to claim in complex RL settings. "Appropriate" or "Adaptive" are indeed more rigorous description for this term.
>
> 1. **Terminology Revision.**
>    We revise the manuscript to describe the mechanism as **"Context-Aware Adaptive Selection"** rather than "Optimal Selection". This terminology more accurately reflects the model's behavior: it does not necessarily find the optimal perspective, but rather **adaptively selecting the appropriate instruction perspective based on the context**. The model learns this through the RL training process to maximize the reward from multiple rollouts.
>
> 2. **Empirical Evidence of Adaptability.**
>    To substantiate that the selection is indeed "adaptive" (and effective) rather than random, Table 7 of the paper illustrate that UI-Ins is capable to select different instruction perspecvtives on different samples. We further conduct a qualitative analysis of reasoning traces and find correlations between UI-element features and reasoning types. For example, for picture-like buttons the model tends to use appearance-based reasoning, whereas for IDE interfaces it tends to use location-based reasoning.
>
> ---
>
> **[Q2] Qualitative Analysis of Emergent Reasoning**
>
> We will expand the qualitative analysis to illustrate *how* the model synthesizes novel paths in our paper through example cases in the revised paper. For example, the UI element is a center align button in Office Software, the different emergent reasoning can be as follows:
>
> ```
> Single Perspective Reasoning:
> Click the button to center the text [Functionality].
>
> Combination Perspectives Reasoning:
> Click the button with the icon of centered horizontal lines [Appearance] to center the text [Functionality].
>
> Combination Perspectives and Emergent Perspective Reasoning:
> To the right of the currently highlighted option [Location], click the inactive button [State], identifiable by its icon of centrally-aligned staggered lines [Appearance], to center the text [functionality], which will set it as the new exclusive active state [Prediction] in alignment control group [Group Affiliation].
> ```
>
> Besides, we also provide a much more detailed statistics analysis of emergent capabilities, and some cases of comparison between UI-Ins with other baselines.
>
>
>
> ---
>
> **Reference**
>
> [1] Schulman, John, et al. "Proximal policy optimization algorithms." arXiv preprint arXiv:1707.06347 (2017).
>
> [2] Rafailov, Rafael, et al. "Direct preference optimization: Your language model is secretly a reward model." Advances in neural information processing systems 36 (2023): 53728-53741.
>
> [3] Ahmadian, Arash, et al. "Back to basics: Revisiting reinforce style optimization for learning from human feedback in llms." arXiv preprint arXiv:2402.14740 (2024).
>
> [4] Yu, Qiying, et al. "Dapo: An open-source llm reinforcement learning system at scale." arXiv preprint arXiv:2503.14476 (2025).

---

### Official Review · Reviewer_iJUH · 2025-10-31

**Soundness:** 4
**Presentation:** 4
**Contribution:** 3
**Rating:** 6
**Confidence:** 4

**Summary:**

This paper is well-written. It addresses the task of GUI grounding. The authors make two primary observations:

1. Existing GUI grounding datasets suffer from a high flaw rate (a manually verified 23.3%).
2. Current models fail to leverage "instruction diversity," the concept that a single intent can be expressed from multiple perspectives (e.g., by appearance, function, or location). The authors demonstrate that simply exploiting this diversity at inference time can yield up to a 76% relative performance improvement.

Based on these insights, the paper introduces the "Instruction as Reasoning" (IR) paradigm, which treats instructions not as static inputs but as dynamic, selectable "analytical pathways". To implement this, the authors propose a high-fidelity data pipeline to clean existing datasets and augment them with multi-perspective instructions. They then introduce a two-stage training framework. The resulting models, UI-Ins-7B and UI-Ins-32B, achieve new SOTA performance on five challenging benchmarks.

**Strengths:**

- The paper presents solid empirical foundation. The preliminary analyses in Section 2 are data-driven conclusions from well-designed controlled experiments. The solid investigation provides an irrefutable "why" for the rest of the paper.
- The "Instruction as Reasoning" (IR) paradigm is a key conceptual contribution. It moves beyond generic "think step-by-step" reasoning and proposes a *structured* form of reasoning that is directly derived from the problem's multimodal context.
- The SFT+RL two-stage framework is an elegant solution. It correctly identifies that the model must first learn *how* to perform a new skill (SFT for reasoning generation) before it can learn *when* to apply it (RL for reasoning selection) .
- The method is not just elegant; it is highly effective. It achieves SOTA on five distinct grounding benchmarks, with particularly strong gains on "advanced" and "implicit" instructions, indicating a deeper semantic understanding. The SOTA on AndroidWorld proves that these offline grounding gains translate directly to improved online agent performance.
- A significant finding is the model's emergent ability to synthesize multiple perspectives and generate entirely new analytical angles not seen in the SFT data. This suggests the SFT+RL framework doesn't just teach the model to memorize four perspectives but encourages a more general and flexible reasoning capability.

**Weaknesses:**

- In 3.2 Multi-Perspective Instruction Augmentation, you mention that you leverage GPT-4.1 (OpenAI, 2025) to generate new instructions from the four fundamental analytical perspectives identified: appearance, functionality, location, and intent. However, you haven’t presented further analysis on the crafted dataset. The paper lacks a systematic comparison between the original dataset and the augmented one to prove the effectiveness.
- This paper only conducts experiments on one online agent benchmark, especially in the context of Android. This might be insufficient to prove the performance of the UI-Ins in dynamic GUI environments

**Questions:**

- Have you explored any other online agentic benchmark and applied your model on different platforms (e.g., Web/Desktop, etc.)?
- Could you share more details of the rationale behind the design of the four components (appearance, functionality, location, and intent) during the data augmentation? Have you done any ablation studies on different components and evaluated their significance?

---

> ### Author Response · Authors · 2025-11-24
> **Author response (1/2)**
>
> **Dear Reviewer iJUH:**
>
> We are encouraged by your comprehensive review and positive assessment. We deeply appreciate your recognition of our paper, including **conceptual novelty in Instruction-as-Reasoning" (IR) paradigm**, **methodological elegance in our SFT+RL framework**, **solid empirical foundation in the preliminary analysis**, **SOTA performance on comprehensive benchmarks**, and the **significant finding in model’s emergent ability**
>
> During rebuttal we have carefully responded to each of your valuable feedback, we **summarize our responses below**:
>
> 1. **Comprehensive comparison between original dataset with augmented dataset. (W1)**
> 2. **Comprehensive systematic analysis of different instruction types. (Q2)**
> 3. **Additional experiments for online benchmarks (OS-World), a comparison with base model and an further error analysis on AndroidWorld. (W2, Q1)**
>
> In light of your insightful comment, we are encouraged to find that these experiments and analysis have largely strengthen our paper. We thank you for your constructive feedback. And we show details in the following response.
>
> ---
>
> __[W1] Systematic comparison between the original dataset and the augmented dataset__
>
> To make a clear comparison between the original dataset and the augmented dataset, we demonstrate as follows:
>
> **1. Augmented data is much more richer in Semantic level**
>
> We present the detailed number of our SFT and RL datasets in **Table R1** and **Table R2** below to show the semantic richness of augmented dataset. Most of the samples have multiple instruction perspectives, which is much more diverse than the original dataset.
>
> __Table R1. Details of SFT data__
>
> |Dataset Name|#Samples|Origin|Appearance|Functionality|Location|Intent|
> |---|---|---|---|---|---|---|
> |AgentNet ubuntu subset|58278|41752|55737|54873|54192|44141|
> |Aria-UI web subset|64826|15665|60194|53520|62988|33884|
> |OSAltas AMEX subset|70023|34215|66255|61948|68075|41157|
> |AgentNet win&mac subset|80357|58713|77122|76151|76522|64298|
> |OSAltas desktop_macos subset|647|441|550|564|615|416|
> |OSAltas desktop_linux subset|832|542|720|730|786|559|
> |OSAltas desktop_windows subset|7030|4298|6210|6229|6671|4450|
> |android_control|8001|5781|7476|7530|7811|4699|
> |Omniact|1159|431|1075|1055|1095|744|
>
> __Table R2. Details of RL data__
>
> |Setting|#Samples|Original|Appearance|Functionality|Location|Intent|
> |---|---|---|---|---|--|--|
> |AgentNet ubuntu subset|4537|3799|2979|4029|3188|3786|
> |AgentNet win&mac subset|8033|5878|5717|6955|5862|6239|
> |Omniact|1520|119|123|148|108|161|
> |OSAltas desktop_windows subset|4651|3303|2888|4139|3180|3985|
> |OSAltas desktop_macos subset|522|347|363|478|392|453|
> |AMEX|6553|3712|5258|5739|5543|4311|
> |OSAltas desktop_linux subset|517|380|325|462|334|453|
> |Aria-UI web subset|5358|1792|3500|3957|4043|2912|
> |android_control|1160|948|980|1110|1030|895|
>
> **2. Augmented dataset effectively improves performance after training**
>
> To provide a more clear comparison of the data quality, we trained augmented data and original dataset separately, as shown in **Table R3**, augmented one significantly improved the performance.
>
> **Table R3. Performance Comparison by training data**
> |Training Data|MM.|I2E.|SS.Pro.|SS.V2|
> |-|-|-|-|-|
> |Original dataset|72.3|63.5|33.0|88.1|
> |Augmented dataset|**76.7**|**70.4**|**40.9**|**90.8**|
>
> ---

---

> ### Author Response · Authors · 2025-11-24
> **Author response (2/2)**
>
> __[W2 & Q1] More Online Benchmark Evaluation:__
>
> We thank the reviewer for pointing out the need for more online evaluation. We expand our evaluation on **OSWorld** (Desktop), and performed a granular failure analysis on AndroidWorld to pinpoint exactly which capabilities are enhanced.
>
> **1. Strong performance on OS-World (Desktop Domain).**
>
> As shown in **Table R4**, UI-Ins-7B achieves a **42.8% Success Rate (SR)** (No A11y tree, additional coding-based actions and multiple rollouts),surpassing famous Agentic Framework like Agent S2.5 w/o3 (39.0%). This confirms that UI-Ins serves as a highly effective executor for generalist agents beyond mobile platforms.
>
> **Table R4. Performance Comparison on OS-World**
> |Model|Type|Max Step|SR|
> |-|-|-|-|
> |opencua-72b-preview [1]|Specialized model|15|39.0|
> |agent s2 [2] w/ gemini-2.5-pro|Agentic Framework|15|34.6|
> |agent s2.5 w/ o3|Agentic Framework|15|39.0|
> |**UI-Ins-7B w/ o3 (Ours)**|Agentic Framework|15|42.3|
> |**UI-Ins-7B w/ GPT-5 (Ours)**|Agentic Framework|15|**42.8**|
>
> **2.Additional In-Depth Analysis on Android World**
>
> To further demonstrate the UI-Ins's effectiveness on online benchmarks, we also conduct additional in-depth analysis on Android World.
> + **Significant improvement compared with base model in Android World.**
>
>    In Android World, We evaluate the base model Qwen2.5-VL-7B w/GPT-5, which can only achieve a **50.0%** SR **vs. 74.1 by UI-Ins-7B** (we fixed the environmental problems and achieve 74.1% SR, which will update in revision), shows the significant improvement of UI-Ins-7B.
>
> + **Failure analysis on Android World**
>
>    To further investigate the robustness of UI-Ins-7B, we manually analyzed failure cases in AndroidWorld. The results in **Table R5** reveal a striking insight:
>    *   **Grounding Solved well by UI-Ins-7B:** Out of all failed trajectories, only 1 case was due to grounding failure (incorrect click).
>    *   **Planning ability becomes the main problem in future research:** The vast majority of failures stemmed from Planning (17) or Step Limitations (11).
>
> **Table R5. Failure Cause Analysis on AndroidWorld**
>    |Failure Type|Count|Percentage|Implication|
>    |-|-|-|-|
>    |Planning Failure|17|58.6%|Planner Hallucination / Loop|
>    |Step Limitation|11|37.9%|Task too long / Inefficient path|
>    |**Grounding Failure**|**1**|**3.5%**|**UI-Ins Precision is robust**|
>    |**Total Failures**|**29**| **100%** ||                              |
>
> ---
>
> __[Q2] Rationale of different instruction type selection and ablation of their significance.__
>
> **1. Rationale of different instruction types selection**
>
> + **Human-like Reasoning:** Our design mimics human cognitive strategies.
> + **Preliminary analysis shows the impact of different instruction types:** we conduct experiments (**Table R6**) reveal that distinct perspectives target different challenges.
>    + Appearance & Functionality: Low-Level visual information, performs well in all three benchmarks.
>    + Intent & Location: High-level information, they not work on SS.Pro and MMBench-GUI L2, but work quite well on UI-I2E Bench, which much more difficult on query understanding, shows the potential impact of these instruction types in complex GUI scenarios.
>
> **Table R6. Preliminary analysis of instruction diversity.**
> | Benchmark| original|App.|Func.|Loc.|Intent|**Combination (Upper Bound)**|
> |-|-|-|-|-|-|-|
> |ScreenSpot-Pro|24.4|35.2|29.8|21.3|26.1|**43.1 (↑76.6%)**|
> |UI-I2E-Bench|56.0|81.8|75.0|70.6|64.1|**86.9 (↑55.2%)**|
> |MMBench-GUI|63.4|79.5|70.9|61.2|63.6|**84.8 (↑33.8%)**|
>
> **2. Systematic analysis of different instruction types: all instruction types have their own benefit.**
>
> + **"Keep-One-Left" Ablation:** we SFT model only in a single instruction type, as shown in **Table R7**.
>
> __Table R7. Performance comparison after SFT solely on single instruction type.__
> |Instruction Set|MM|I2E|Show|SS.Pro.|SS.V2|
> |-|-|-|-|-|-|
> |App.|76.0|69.1|65.5|**37.4**|**91.2**|
> |Func.|75.4|69.1|64.1|**37.4**|90.6|
> |Loc.|74.2|64.9|62.7|37.1|90.0|
> |Intent|73.9|67.6|63.4|32.6|88.8|
> |**Full(Ours)**|**76.7**|**70.4**|**66.6**|**40.9**|90.8|
> + **"Leave-One-Out" Ablation:** we SFT model except each single instruction type, as shown in **Table R8.**
>
> __Table R8. Leave-one-out ablation study of different component__
> |Instruction Set|MM|I2E|Show|SS.Pro.|SS.V2|
> |-|-|-|-|-|-|
> |w/o App.|**76.7**|69.0|**67.1**|39.2|89.6|
> |w/o Func.|75.9|69.1|63.7|39.3|90.7|
> |w/o Loc.|76.6|68.8|66.6|39.7|89.6|
> |w/o Intent|**76.7**|68.1|66.4|40.6|**91.3**|
> |**Full (Ours)**|**76.7**|**70.4**|66.6|**40.9**|90.8|
>
> Results from **Table R7** and **Table R8** show all types can help in grounding, and full set can achieve a more balanced performance.
>
> ---
> **Reference**
>
> [1] Wang, Xinyuan, et al. "Opencua: Open foundations for computer-use agents." arXiv preprint arXiv:2508.09123 (2025).
>
> [2] Agashe, Saaket, et al. "Agent s2: A compositional generalist-specialist framework for computer use agents." arXiv preprint arXiv:2504.00906 (2025).

---

### Official Review · Reviewer_7NYR · 2025-11-01

**Soundness:** 3
**Presentation:** 2
**Contribution:** 3
**Rating:** 6
**Confidence:** 5

**Summary:**

This paper aims to address the lack of diversity in reasoning for GUI grounding. It proposes a data synthesis pipeline and a two-stage training framework. The resulting model achieves strong performance on grounding tasks and improves the accuracy of existing methods in real-world navigation scenarios.

**Strengths:**

1. This paper offers a novel perspective: instruction and reasoning diversity.
2. The overall presentation of the paper is fairly clear.
3. The experimental section is comprehensive.

**Weaknesses:**

1. One of the core contributions of the paper is the construction of a data synthesis pipeline. However, the description in Section 3 remains unclear:

    1.1 There is a lack of description of the raw dataset formats. Although Tables 9 and 10 in the Appendix describe the datasets used for training, they do not provide the original dataset details. For example, what is the original size of OSAtlas, and which subset of OSAtlas is used? In addition, the dataset meta information though full pipeline is unclear. For example, how to process the multi-step dataset AgentNet. How much data is remained at each stage.


    1.2 The formalization of the training data synthesis pipeline is missing. Since most grounding datasets consist of a single image paired with multiple element annotations, the statement on Line 215, “For each data instance, the model receives the screenshot with the highlight,” is ambiguous. Does “data instance” refer to one image together with its set of element annotations, or to a single element?

2. The analysis of the online evaluation experiments is relatively weak. While the overall performance improves, it is unclear which specific capabilities are enhanced; more detailed analysis is needed.

**Questions:**

1. In Line 918, is the Action Space in the SFT training prompt fixed to only one action?

2. The paper states that during the SFT stage, the diversified instruction serves as the reasoning. In the actual SFT training data, is the reasoning directly the diversified instruction? If so, does the instruction then need to be replaced with a standard instruction? The construction process of the data samples is not sufficiently clear.

3. In the data synthesis pipeline, the diversified instruction is used as reasoning. Is the instruction used for training also augmented accordingly?

4. AgentNet is a multi-step trajectory dataset that includes many non-grounding actions. How is this type of dataset converted into the Grounding format?

5. Sampling of SFT data: According to Equation (1), the final reasoning used for training is sampled from four perspectives. Why not use the full dataset directly for training?

---

> ### Author Response · Authors · 2025-11-24
> **Author response (1/3)**
>
> **Dear reviewer 7NYR:**
>
>
> We sincerely thank Reviewer 7NYR for the constructive feedback and positive assessment. We are particularly encouraged by your recognition of our **novel perspective on instruction and reasoning diversity**, as well as your appreciation for the **clarity of our presentation** and the **comprehensiveness of our experimental results**.
>
> We deeply appreciate your constructive feedback, which help us improved the work:
>
> * **New SOTA by using the full augmented dataset (Q5).**
> * **Further in-depth analysis on Android World, and one more online benchmark evaluation (OS-World) (W2).**
> * **Training with more GUI actions proves the robustness of our method (Q1).**
> * **Presentation improvement (W1).**
>
> Below, we provide detailed responses for all your valuable comments:
>
>
> ---
>
> __[W1 & Q2 & Q3 & Q4] Unclear statement about data processing pipeline:__
>
>
> **1. [W1.1 & Q4] Data Pipeline Transparency: Processing Logic and Statistics**
>
>  We thank the reviewer for highlighting the need for better transparency. We will add a detailed "Dataset" section in the Appendix. Below, we clarify the specific processing logic for raw datasets and AgentNet.
>
> + **Raw Dataset Refinement (IoU Strategy).**
>   To rectify annotation noise in large-scale datasets (e.g., OS-Atlas), we employ OmniParser V2 for spatial refinement. We calculate the IoU between the original ground truth (GT) and detected UI elements; if the maximum match exceeds a threshold, we refine the GT to the intersection of the original box and the detected element. This ensures the annotation tightly bounds the actual visual target.
>
> + **Converting AgentNet Trajectories to Grounding Format**
>
>   >  __[Q4]Convert AgentNet to Grounding format:__
>
>   We process AgentNet by decomposing multi-step trajectories into single steps and retaining only click-related actions. The metadata's "thought" field serves as the instruction. For spatial labels, we map the raw click coordinates to multiple bounding boxe s by identifying the enclosing UI elements detected by OmniParser V2 and computing their spatial intersection.
>
> + **Data Statistics.**
>
>   >  __[Q2 & Q3] Instruction Construction Strategy.__
>
>   We present the detailed composition of our SFT and RL datasets in **Table R1** and **Table R2** below.
>
>      *   **SFT Stage (Teacher-Forcing):** The input $I_{in}$ is selected from the augmented set $S$. The reasoning label is a **distinct** instruction $I_{reason} \in S \setminus \{I_{in}\}$. This explicitly teaches the model to map between different perspectives. *Note: We now utilize full-dataset training for sampling.*
>
>      *   **RL Stage (Adaptive selecting):** We provide only the input $I_{in}$. The model must generate the reasoning without GT supervision, allowing it to dynamically explore and select the reasoning perspective that maximizes the reward.
>
>
>    __Table R1. Details of SFT data__
>
>    |Dataset Name|#Samples|Origin|Appearance|Functionality|Location|Intent|
>    |---|---|---|---|---|---|---|
>    |AgentNet ubuntu subset|58278|41752|55737|54873|54192|44141|
>    |Aria-UI web subset|64826|15665|60194|53520|62988|33884|
>    |OSAltas AMEX subset|70023|34215|66255|61948|68075|41157|
>    |AgentNet win&mac subset|80357|58713|77122|76151|76522|64298|
>    |OSAltas desktop_macos subset|647|441|550|564|615|416|
>    |OSAltas desktop_linux subset|832|542|720|730|786|559|
>    |OSAltas desktop_windows subset|7030|4298|6210|6229|6671|4450|
>    |android_control|8001|5781|7476|7530|7811|4699|
>    |Omniact|1159|431|1075|1055|1095|744|
>
>    __Table R2. Details of RL data__
>
>    |Setting|#Samples|Original|Appearance|Functionality|Location|Intent|
>    |---|---|---|---|---|--|--|
>    |AgentNet ubuntu subset|4537|3799|2979|4029|3188|3786|
>    |AgentNet win&mac subset|8033|5878|5717|6955|5862|6239|
>    |Omniact|1520|119|123|148|108|161|
>    |OSAltas desktop_windows subset|4651|3303|2888|4139|3180|3985|
>    |OSAltas desktop_macos subset|522|347|363|478|392|453|
>    |AMEX|6553|3712|5258|5739|5543|4311|
>    |OSAltas desktop_linux subset|517|380|325|462|334|453|
>    |Aria-UI web subset|5358|1792|3500|3957|4043|2912|
>    |android_control|1160|948|980|1110|1030|895|
>
> **2. [W1.2] Pipeline Formalization & Instance Definition.**
>
> + **Instance Definition:** A "Data Instance" refers to a **single target UI element**. A screenshot containing $N$ elements is treated as $N$ independent instances.
>
> *   **Pipeline:** For a target box $\mathbf{b}_{gt}$ in screenshot $\mathbf{S}$:
>
>     + **Visual Highlighting:** Generate $\mathbf{S}' = \text{Overlay}(\mathbf{S}, \mathbf{b}_{gt})$ with a visual marker to focus the UI element.
>
>     + **Generation:** Query GPT-4.1 to generate instructions $I_{gen} = \{I_{app}, I_{func}, I_{loc}, I_{intent}\}$ based on $\mathbf{S}'$ and $I_{ori}$.
>
>     + **Verification:** Filter $I_{gen}$ by verifying if each generated instruction unambiguously maps back to $\mathbf{b}_{gt}$ in $\mathbf{S}'$.

---

> ### Author Response · Authors · 2025-11-24
> **Author response (2/3)**
>
> **[W2] Further Analysis of Online Evaluation:**
>
> We thank the reviewer for pointing out the need for deeper online analysis. We have provide a comparison between UI-Ins-7B  with its base model Qwen2.5-VL-7B , except this, we conduct a failure analysis on Android World to show UI-Ins-7B sloved grounding well in dynamic environment, furthermore, we expanded our evaluation on the challenging **OS-World** benchmark and achieves a **42.8%** success rate.
>
> 1. **Significant improvement compared with Base model.**
>
>    In Android World, we evaluate the base model Qwen2.5-VL-7B as the executor and GPT-5 as planner, which can only achieve a **50.0%** success rate, however, UI-Ins-7B only trained with grounding data, can easily achieve a success rate at **74.1%** (we fixed the environmental problems and achieve 74.1% success rate, which will update in revision), illustrating the effectiveness of UI-Ins-7B in online evaluation.
>
> 2. **Failure analysis on Android World**
>
>    The reviewer asked *"which specific capabilities are enhanced."* To answer this, we manually analyzed failure cases in AndroidWorld. The results in **Table R3** reveal a striking insight:
>
>    *   **Grounding Solved well by UI-Ins-7B:** Out of all failed trajectories, only 1 case was due to grounding failure (incorrect click). The improvement in grounding capability is the main reason for the significant gains compared to the base model.
>
>    *   **Planning ability becomes the main problem in future research:** The vast majority of failures stemmed from Planning (17) or Step Limitations (11).
>
>    **Table R3. Failure Cause Analysis on AndroidWorld**
>
>    |Failure Type|Count|Percentage|Implication|
>    |:--|:--|:--|:--|
>    |Planning Failure|17|58.6%|Planner Hallucination / Loop|
>    |Step Limitation|11|37.9%|Task too long / Inefficient path|
>    |**Grounding Failure**|**1**|**3.5%**|**UI-Ins Precision is robust**|
>    |**Total Failures**|**29**| **100%** ||
>
> 3. **Further evaluation: great performance on OS-World (Desktop Domain).**
>
>    To further investigate the robustness of our method, we further evaluate the famous online benchmark OS-World, as shown in **Table R4**, UI-Ins-7B achieves a **42.8%** Success Rate (No A11y tree, additional coding-based actions and multiple rollouts), surpassing famous Agentic Framework like agent S2.5 (39.0%). This confirms that UI-Ins serves as a highly effective executor for generalist agents beyond mobile platforms.
>
>    **Table R4. Performance Comparison on OS-World**
>
>    | Model|Type| Max Step | Success Rate |
>    |--|--|--|--|
>    |opencua-72b-preview [1]|Specialized model|15|39.0|
>    |agent s2 [2] w/ gemini-2.5-pro|Agentic Framework|15|34.6|
>    |agent s2.5 w/ o3|Agentic Framework|15|39.0|
>    |**UI-Ins-7B (o3 as planner) (Ours)**|Agentic Framework|15|42.3|
>    |**UI-Ins-7B (GPT-5 as planner) (Ours)**|Agentic Framework|15|**42.8**|
>
>
>
>
>
> ---
>
>
> **[Q1] Is the action space in the SFT training prompt fixed to only one action?**
>
> 1. **Clarification on Main Paper Setting.**
>    Yes, for our specialized grounding models, the SFT action space is fixed to coordinate prediction.
>
> 2. **Extension: Training with more GUI actions.**
>    We agree with the reviewer that real-world agents require a richer action space. Inspired by this question, we trained a Generalist Agent with both **Mobile data in complex action space** and **Grounding data in single action space**  to verify if our IR paradigm benefits complex action planning. The results shown in **Table R5** demonstrate our method can contribute to better agent performance on Android Control, and IR-RL can keep the generalist agent performance even trained on only grounding data.
>
>    **Table R5. IR's impact together with Mobile data(Complex action space)**
>
>    |Row|Training method|Training data type|Android Control High|Android Control Low|MMBench-GUI L2|UI-I2E|Showdown|ScreenSpot-Pro|ScreenSpot-V2|
>    |---|---|---|--|--|--|--|--|--|--|
>    |1|-|-|62.9|85.0|63.4|56.0|43.6|22.4|86.5|
>    |2|SFT|Mobile|72.7|88.1|-|-|-|-|-|
>    |3|SFT|Mobile + Grounding|74.2|90.4|76.0|69.7|64.3|42.2|89.7|
>    |4|+RL|Grounding|**74.7**|90.2|**82.8**|**81.2**|**71.6**|**53.6**|**93.3**|
>
> **Conclusion:** **Instruction-as-Reasoning serves as a fundamental capability enhancer**.
>
> ---

---

> ### Author Response · Authors · 2025-11-24
> **Author response (3/3)**
>
> **[Q5] Why not use the full augmented dataset for training? (New SOTA)**
>
> 1. **Valuable Insight: Full Data Training Yields New SOTA.**
>    We sincerely thank the reviewer for this constructive suggestion. Following your advice, we trained on full data, and achieve a new SOTA performace, results are shown in **Table R6**
>
>    __Table R6. Full instructions outperformed Random one selection strategy__
>
>    |Method|data|MMBench-GUI L2|UI-I2E|Showdown|ScreenSpot-Pro|ScreenSpot-V2|
>    |---|---|---|--|-|-|-|
>    |SFT Random 1|29w for SFT|76.3|70.1|**67.5**|37.1|90.6|
>    |SFT All| 117w for SFT |**76.7**|**70.4**|66.6|**40.9**|**90.8**|
>    |SFT Random 1 + RL|10w for RL|83.1|81.1|73.1|52.2|94.0|
>    |SFT All + RL|10w for RL|**83.5**|**82.4**|**73.6**|**53.9**|**94.1**|
>
> 2. **Rationale for Initial Sampling Strategy.**
>    Our original decision to use random sampling was driven by two considerations for experimental rigor:
>
>    *   **Strict Ablation Fairness:** We aimed to keep the SFT data size comparable to the baselines to ensure that gains were attributed to the *IR* rather than simply *more data*.
>    *   **Overfitting Concerns:** We were initially cautious that exposing the model to the same screenshot multiple times (with different text) might cause it to memorize coordinate patterns ("overfitting to the image").
>
>
>
> ---
>
> __The Core novelty and contributions of our work.__
>
> We wish to take the chance to highlight the core algorithmic and theoretical contributions of this work, which go beyond data pipeline:
>
> 1. **Systematic analysis of Instruction in GUI grounding**, including quality and diversity, support the community and explained   "Why" in our paper.
> 2. **Instruction-as-Reasoning Paradigm:** allowing model use the adaptive reasoning pathway in different scenarios.
> 3. **SOTA performance on 5 grounding benchmarks and online benchmark AndroidWorld**, also shows the competitive performance on OS-World.
> 4. In-depth analysis:
>    + Instruction-as-Reasoning address the critical issue that **how reasoning can be formulated to enhance, rather than hinder grounding performance**.
>    + Instruction-as-Reasoning **mitigates policy collapse** in the SFT+RL framework in End2End GUI grounding task.
>    + Instruction-as-Reasoning **unlocks emergent reasoning capabilities**, allowing the model to reason from novel perspectives.
>
> ---
>
> **Reference**
>
> [1] Wang, Xinyuan, et al. "Opencua: Open foundations for computer-use agents." arXiv preprint arXiv:2508.09123 (2025).
>
> [2] Agashe, Saaket, et al. "Agent s2: A compositional generalist-specialist framework for computer use agents." arXiv preprint arXiv:2504.00906 (2025).

---

### Official Review · Reviewer_zWAk · 2025-11-02

**Soundness:** 2
**Presentation:** 2
**Contribution:** 2
**Rating:** 2
**Confidence:** 4

**Summary:**

This paper first identifies two types of flaws in current GUI grounding datasets: insufficient prompt diversity and poor prompt quality. The authors conduct preliminary experiments showing that pairing a model with the instruction perspective most suitable for a given task can boost grounding performance even without retraining. They then develop a data-processing framework to filter out unclear instructions and increase perspective diversity, and introduce "Instruction-as-Reasoning," a two-stage training pipeline to further improve grounding. Experiments demonstrate new state-of-the-art results on five major grounding benchmarks.

**Strengths:**

1. New SOTA performance across five grounding benchmarks
2. Propose improving grounding ability by matching the most suitable instruction type to each task.

**Weaknesses:**

1. I am doubtful about the contribution of the "Instruction as Reasoning" paradigm. I think this paradigm is essentially just a combination of "instruction augmentation + chain-of-thought generation". Compared to existing multi-perspective instruction augmentations (such as Jedi, UI-E2I-Synth, etc.), does this work possess genuine novelty at the algorithmic level, or is it merely an innovation in framework naming?
2. Also, the paper claims "instructions as dynamic reasoning pathways," but in practice, it primarily employs a two-stage training mechanism of SFT+GRPO. What is the fundamental difference between this mechanism and the typical RLHF/GRPO structure? Is it simply a renaming of "prompt diversity" to "reasoning diversity"?
3. The figure and the table in this paper are not perfect and lack inspiration for the readers. For example, Figure 6 is unclear and should be either condensed or supplemented with more useful information. I am interested in how the model selects optimal strategies across different perspectives during RL, which the figure does not show. Researchers in the field are likely already familiar with the "Advantages" and "Objectives" sections, making them redundant.

**Questions:**

1. Do all baselines use the same data cleaning and preprocessing? The paper points out that the original data has a 23.3% defect rate. If the baseline model is trained on uncleaned data, while this method uses cleaned and augmented data, the performance improvement may come from the difference in data quality rather than algorithm improvement.
2. When related to "ambiguous & mismatch", did the authors measure human–AI agreement?
3. In Figure 6, if the author could choose a clearer screenshot of the input image to explain, that would be better.

---

> ### Author Response · Authors · 2025-11-24
> **Author response (1/3)**
>
> **Dear Reviewer zWAk:**
>
>   We sincerely thank Reviewer zWAk for the feedback. We appreciate your recognition of our **new SOTA performance across five major benchmarks** and the effectiveness of our core insight: **improving grounding ability by matching the most suitable instruction type to each task**. However, regarding the concerns about novelty and the contributions, we clarify them as follows and we will update into our revised paper to make these points more clear.
>
>   ---
>
>   __[W1] Novelty and Contributions of "Instruction-as-Reasoning"__
>
>   We respectfully clarify that **Instruction-as-Reasoning (IR) solving the critical issue in grounding that standard CoT reasoning fails** and it is not merely a combination of augmentation and CoT, but a novel paradigm designed to mimic human cognitive strategies. In light of your feedback, we will revise the paper to highlight the novelty and contribution of IR.
>
>   **1. Standard CoT often degrades end-to-end GUI grounding performance,  IR effectively makes reasoning works for grounding.** Prior works (e.g., UI-R1 [1], GUI-G2 [2], GTA1 [3], GUI-G1 [4]) and our own experiments (in Sec. 4.4) reveal that standard CoT reasoning (which we term Free-Form Reasoning) consistently harms GUI grounding accuracy as shown in **Table R1**, whereasour IR approach consistently improved the performance across benchmarks.
>
>   > GUI-G1: Thinking leads to poorer grounding performance
>
>   **Table R1. Comparison of Free-Form Reasoning (FFR) vs. Instruction-as-Reasoning (IR)**
>
> |Source| Method|Reasoning Type| ScreenSpot-Pro| ScreenSpot-V2|
> |---|---|---|---|---|
> |UI-R1|UI-R1| - | 32.1|-|
> ||UI-R1|FFR|17.8 (&darr;44.5%)|-|
> |GUI-G2|GUI-G2 |-|-|93.3|
> ||GUI-G2|FFR|-|88.7(&darr;4.9%)|
> |GTA1|GTA1 (Base on UI-Tars-1.5 7B)|-|50.1|92.4|
> ||GTA1 (Base on UI-Tars-1.5 7B)|FFR|46.9 (&darr;6.4%)|93.2 (&darr;0.8%)|
> |Ours Validation on FFR|Ours validation Exp on Qwen2.5-VL-7B|-|36.4|91.7|
> ||Ours validation Exp on Qwen2.5-VL-7B|FFR|36.4(&darr;0.0%)|91.6(&darr;0.1%)|
> |Ours Validation on IR|Ours validation Exp on UI-Tars-1.5 7B|-|48.7|91.7|
> ||Ours validation Exp on UI-Tars-1.5 7B|IR|**51.2(&uarr;5.1%)**|**91.9(&uarr;0.2%)**|
> ||**UI-Ins-7B (Ours)**|-|47.5|93.1|
> ||**UI-Ins-7B (Ours)**|IR|**52.2(&uarr;9.9%)**|**94.0(&uarr;0.9%)**|
>
>
>
>   **2. Inference with human-like cognitive strategies.** The critical question is: how can we make reasoning beneficial to grounding? Unlike prior works, such as JEDI or UI-E2I-Synth, that treat diverse instructions as input queries to improve generalization performance, our proposed IR trains models to actively reason over them at inference stage. This training process enhances model reasoning ability for GUI grounding with human-like cognitive strategies. As shown in **Table R2** (Sec. 2.1 in paper), distinct perspectives influenced grounding performance without training, and model performance can significantly improved if they can always select the appropriate perspective.
>
>   **Table R2. Preliminary analysis of instruction diversity and the potential of dynamic selection.**
>   | Benchmark| riginal|App.|Func.|Loc.|Intent| **Combination (Upper Bound)**|
>   |---|---|---|---|---|---|---|
>   | ScreenSpot-Pro|24.4|35.2|29.8|21.3| 26.1  |**43.1 (↑76.6%)**|
>   |UI-I2E-Bench|56.0|81.8|75.0|70.6|64.1|**86.9 (↑55.2%)**|
>   |MMBench-GUI|63.4|79.5|70.9|61.2|63.6|**84.8 (↑33.8%)**|
>
> In summary, IR solves a key challenge: direct CoT reasoning fails n grounding. It address this by employing a novel human-like cognitive strategy. We are encouraged that reviewer iJUH highlighted the IR paradigm as a **"key conceptual contribution"** and Reviewer UPmG described it as **"novel"**.
>
> ---

---

> ### Author Response · Authors · 2025-11-24
> **Author response (2/3)**
>
> __[W2] The Novelty of SFT+RL Framework in IR: Solving the Policy Collapse in Grounding.__
>
> In light of your valuable feedback, we will highlight the contribution of our SFT+RL paradigm more clear. Specifically, **IR solved an important problem in SFT+RL paradigm in GUI grounding: Policy Collapse.** Additionally, **IR is fundamentally distinct from "prompt diversity" and it's robustness even scaling data in SFT.**
>
>
>
>
> **1. An Important Issue: SFT+RL fails in grounding training (Policy Collapse) `even using rich prompt diversity`**
>
>   As demonstrated by Phi-Ground [5] and our own experiments (Sec. 4.4 of paper), applying RL (PPO/GRPO) directly to standard SFT grounding models leads to failure, as shown in **Table R3**: shows SFT models (JEDI, Qwen2.5-VL SFT by us) trained with diverse prompts but without IR still suffer from collapse during RL. This proves that **prompt diversity alone is insufficient to support RL.** However, IR can substantially address the issue of policy collapse in grounding.
>
>   > **Phi-Ground**:
>      i. Although some previous work has shown that RL can provide benefits in purely perceptual tasks, these studies typically begin with relatively low baselines or apply RL directly from scratch without prior SFT.
>      ii. The absence of exploration and reasoning led to low diversity in the answers among rollouts for the same sample, frequently resulting in all rollouts being either entirely correct or entirely incorrect.
>
>   __Table R3. Model meets policy collapse in GUI grounding but IR doesn't.__
>
>   | Experiment|Base Model|Method|ScreenSpot-Pro|ScreenSpot-V2|
>   |---|---|---|---|---|
>   |Experiment on JEDI|JEDI-7B|-|39.5|91.7|
>   ||JEDI-7B|RL|34.5(&darr;12.7%)|83.7(&darr;8.7%)|
>   |Experiment on Qwen-2.5-VL-7B-SFT|Qwen2.5-VL-7B|-|24.4|86.5|
>   ||Qwen2.5-VL-7B|SFT (w/o IR)|37.0|90.6|
>   ||Qwen2.5-VL-7B|SFT + RL (w/o IR)|34.9(&darr;5.7%)|89.9(&darr;0.8%)|
>   |**Experiment of UI-Ins (Ours)**|Qwen2.5-VL-7B|-|24.4| 86.5|
>   ||**Qwen2.5-VL-7B (Ours UI-Ins)**|SFT (w/ IR)|37.1|90.6|
>   ||**Qwen2.5-VL-7B (Ours UI-Ins)**|SFT+RL (w/ IR)|**46.0(&uarr;24.0%)**|**92.8(&uarr;2.4%)**|
>
>
>
>
> **2. Why SFT+RL works by Instruction-as-Reasoning: diverse exploration space**
>
>   Instruction-as-Reasoning forces the model to reason from an explicit instruction perspective before predicting coordinates. This intermediate step expands the action space from "just coordinates" to "instreuction-as-reasoning paths + coordinates." During RL, the model explores different **reasoning paths** (e.g., analyzing Function vs. Intent) to find the correct answer. This diversity in the **reasoning space** prevents the policy from collapsing, as illustrated below:
>
> ```
>   [Rollout 1]
>
>   Reasoning: <think>I will insert a new column into the worksheet.[Function]</think>
>
>   Prediction: [1086, 116] -> Reward: 0.0 (Incorrect)
>
>   [Rollout 2]
>
>   Reasoning: <think>I will add a new cell at the current location in the spreadsheet.[Intent]</think>
>
>   Prediction: [1269, 86] -> Reward: 1.0 (Correct)
>
> ```
>
>
>
> **3. Robustness at Scale: Validation on 1M+ SFT Data.**
>
>   To confirm that IR's effectiveness is not an artifact of small data scales, we expanded our SFT dataset from 290k to 1.17M samples (incorporating advice from Reviewer 7NYR). As shown in **Table R4**, even with massive data scaling, the RL stage with IR continues to yield performance gains, proving that IR is a scalable and essential component for SFT+RL in grounding.
>
>
>
>   __Table R4. When we scaling the SFT data from 29w to 117w, SFT+RL still work by Instruction-as-Reasoning, and benefits the final performance.__
>
>   |Training Method|instruction sampling strategy|data number|MMBench-GUI L2|UI-I2E|Showdown|ScreenSpot-Pro|ScreenSpot-V2|
>   |---|---|---|---|---|---|---|---|
>   |SFT only| Random 1|29w SFT|73.5|68.5|**67.5**|37.1|90.6|
>   ||All instructions|117w SFT|**76.7**|**70.4**|66.6|**40.9**|**90.8**|
>   |SFT + RL|Random 1|10w RL|83.1|81.1|73.1|52.2|94.0|
>   ||All instructions|10w RL|**83.5**|**82.4**|**73.6**|**53.9**|**94.1**|
>
>
>
> **Conclusion: IR is a methodologically novel and solves the critical policy collapse issue in SFT+RL.**
>
> Our contribution lies in identifying that **Reasoning Generation (IR)** enables SFT+RL to work in GUI grounding. It prevents the policy collapse observed in prior works (Phi-Ground) by establishing a structured and diverse exploration space, which is a contribution fundamentally distinct from simply increasing data diversity. We are encouraged that this methodological value was recognized by other reviewers; for instance, Reviewer iJUH found this approach **"elegant"**, and Reviewer UPmG described it as **"methodologically sound"**.

---

> ### Author Response · Authors · 2025-11-24
> **Author response (3/3)**
>
> __[W3 & Q3] Presentation Improvements and Clarification on RL Strategy Selection__
>
> > __[Q3] Screenshot case in Figure 6 is not good enough__
>
> **Clarification: model selects context-aware adaptive strategies during RL by reward maximization.**
>
> The reviewer noted that Figure 6 did not explicitly visualize the selection mechanism. We clarify that the "selection" is learned driven by reward maximization. And we will improve the statement in corresponding figures and paragraphs to make a clearer presentation.
>
>
> ---
>
>
>
>
> __[Q1] The performance improvement comes from the difference in data quality or algorithm improvement?__
>
> **1. Algorithm Improves More than Data Quality**
>
>   To evaluate the improvements from data quality and IR algorithm, we provide a detailed ablation in **Table R5**, here are two mainly conclusions from these results.
>
>   + **Effect of Data Cleaning (Row 1 → Row 2)**: Moving from original noisy data to our cleaned dataset yields a moderate improvement.
>
>   + **Effect of IR Algorithm (Row 2 → Row 4)**: Applying our full SFT+RL pipeline with IR on the same cleaned data yields a massive improvement.
>
>
>
>   **Table R5. Attribution of Performance Gains: Data Quality vs. IR Algorithm**
>
>   | ID      | Data Source   | Training Stage | Reasoning in RL  | MMBench-GUI   | UI-I2E      | ScreenSpot-Pro  | ScreenSpot-V2  |
>   | :----------- | :--------------- | :------------- | :----------------- | :-------------- | :--------------- | :--------------- | :-------------- |
>   | 1      | Original (Noisy) | SFT      | N/A        | 72.3      | 63.5       | 33.0       | 88.1      |
>   | 2      | Cleaned     | SFT      | N/A        | 74.3 (**↑2.0**) | 66.3 (**↑2.8**) | 33.7 (**↑0.7**) | 90.2 (**↑2.1**) |
>   | 3      | Cleaned     | SFT+RL     | No (Direct Coords) | 81.6      | 76.2       | 47.5       | 93.1      |
>   | **4 (Ours)** | Cleaned     | SFT+RL     | **Yes (IR)**    | **83.1 (↑8.8)** | **81.1 (↑14.8)** | **52.2 (↑18.5)** | **94.0 (↑3.8)** |
>
>
> **2. Internal Controlled Experiments Ensure Fair Comparison.**
>
>   Though other baselines like InfiGUI-G1 have different data processing strategies like difficulty filtering, retraining all of them from scratch is computationally prohibitive. However, **Rows 2, 3, and 4 in Table R5 serve as strict internal baselines**.
>
>
>
>
> ---
>
>
>
>
> __[Q2] Measure of human-AI agreement__
>
> We appreciate the reviewer asking for clarification on data reliability. We confirm that the "manual inspection" described in Section 2.2 and Appendix C.2 effectively functions as a rigorous **Human-AI agreement measurement**.
>
> **1. Human-AI Agreement: Validated High Consistency (>93%)**
>
>   As shown in **Table 8**, out of 1,542 samples that the AI claimed were "Precise," human annotators confirmed that **1,443** were indeed "Precise Match," while only 99 were identified as false positives (Ambiguous or Mismatch). This results in a *Human-AI agreement rate of 93.58%, demonstrating that the AI's judgment aligns highly with human ground truth.
>
>
>
>
> **2. Statistical Significance.**
>
>   To verify that this high agreement was not due to chance, we performed a statistical significance test using the **Binomial Distribution**, the observed consistency (93.58%) is statistically significant (p≪0.001). This confirms the reliability of our pipeline in establishing a high-quality data foundation.
>
>
>
> ---
>
>
> Finally, we would like to highlight the **the Novelty and Contributions of UI-Ins**
>
> **1. Systematic analysis of Instruction in GUI grounding**, including quality and diversity, support the community and explained all "Why" in our paper.
>
> **2. Instruction-as-Reasoning Paradigm:** allowing model use the adaptive reasoning pathway in different scenarios.
>
> **3. SOTA performance on 5 grounding benchmarks and online benchmark AndroidWorld**, also shows the competitive performance on OS-World.
>
> **4.** In-depth analysis:
>
>   + **Instruction-as-Reasoning mitigates the challenge that standard reasoning fails** in GUI grounding.
>
>   + Instruction-as-Reasoning **mitigates policy collapse** in the SFT+RL framework in End2End GUI grounding task.
>
>   + Instruction-as-Reasoning **unlocks emergent reasoning capabilities**, allowing the model to reason from novel perspectives.
>
> ---
>
> **Reference**
>
> [1] Lu, Zhengxi, et al. "UI-R1: Enhancing Efficient Action Prediction of GUI Agents by Reinforcement Learning." arXiv preprint arXiv:2503.21620 (2025).
>
> [2] Tang, Fei, et al. "GUI-G $^ 2$: Gaussian Reward Modeling for GUI Grounding." arXiv preprint arXiv:2507.15846 (2025).
>
> [3] Yang, Yan, et al. "Gta1: Gui test-time scaling agent." arXiv preprint arXiv:2507.05791 (2025).
>
> [4] Zhou, Yuqi, et al. "Gui-g1: Understanding r1-zero-like training for visual grounding in gui agents." arXiv preprint arXiv:2505.15810 (2025).
>
> [5] Zhang, Miaosen, et al. "Phi-ground tech report: Advancing perception in gui grounding." arXiv preprint arXiv:2507.23779 (2025).

---

> > ### Comment · Reviewer_zWAk · 2025-11-25
> >
> > Thanks for the authors' clarifications. I now believe that IF does differ from other instruction-augmentation methods and helps RL improve the model's grounding ability. I like Table R5 and hope the authors can incorporate it into the main paper. Also, is the change in Jedi's scores on ScreenSpot-V2 in Table R3 a mistake?
> >
> > To conclude, I agree with the authors and will update my score accordingly.

---

> > > ### Author Response · Authors · 2025-11-27
> > > **Response by Author**
> > >
> > > **Dear Reviewer zWAk:**
> > >
> > > We sincerely thank you for your quick feedback and for acknowledging the distinction and effectiveness of our Instruction-as-Reasoning (IR) paradigm. We are glad to hear that Table R5 proved helpful in resolving your concerns, and we deeply appreciate your time and effort in reviewing our rebuttal, as well as your decision to update the score.
> > > Regarding your specific comments:
> > >
> > > + Incorporating **Table R5**: We fully agree with your suggestion and will add Table R5 into the main paper to clearly show the improvement of IR algorithm and data quality.
> > >
> > > + JEDI Scores in **Table R3**: Thank you for your rigorous review, we apologize for the typo in Table R3, the performance of JEDI-7B (+RL) on ScreenSpot-V2 is 83.7. *Note: To ensure a strictly fair comparison, we re-evaluated JEDI-7B using our implementation and the JEDI-7B's result is near or a slightly higher than the performance reported in original paper. We have corrected our rebuttal now.*
> > >
> > > Thank you again for your keen attention to detail, which has significantly improved the quality of our work.

---

### Author Response · Authors · 2025-11-27
**Global Response**

We sincerely thank the reviewers for the time and effort dedicated to reviewing our paper.

**We are deeply grateful for the consensus on the value of our work:**

*   **Foundational Analysis:** Reviewers praised our systematic analysis of instruction diversity and quality (*zWAk, 7NYR, iJUH, UPmG*), noting it provides a strong motivation and the "irrefutable why" for our approach (*iJUH, UPmG*).

*   **Novelty & Methodology:** The Instruction-as-Reasoning (IR) paradigm is recognized as novel (*zWAk, iJUH, UPmG*), and the SFT+RL framework is deemed elegant and methodologically sound (*iJUH, UPmG*).

*   **Effectiveness of IR:** Reviewers agreed that IR enables the model to select the most appropriate reasoning pathway and unlocks emergent capabilities (*zWAk, iJUH, UPmG*). Crucially, it stabilizes SFT+RL training and succeeds where Free-Form Reasoning (FFR) typically fails (*zWAk, UPmG*).

*   **SOTA Performance & Solid Experiments:** The SOTA results on grounding benchmarks and AndroidWorld were widely acknowledged (*All Reviewers*). The experiments were deemed comprehensive and solid (*7NYR, UPmG*).

*   **Presentation:** The paper was described as well-written (*iJUH*) with clear overall presentation (*7NYR*).

---

**Summary of Experiments and Clarifications in Rebuttal:**

In light to your valuable feedback, we have updated the manuscript with the following experiments:

*   **New Online Benchmark(OS-World):** We expanded the online evaluation to the Desktop domain (OS-World), achieving a competitive **42.8%** success rate.
*   **Expanded Online Analysis (AndroidWorld):**
    *   **Baseline Comparison:** Our UI-Ins-7B achieves a **74.1%** success rate (we fixed the environmental problems and achieve 74.1% success rate, which will update in revision), significantly outperforming the base Qwen2.5-VL-7B (**50.0%**).
    *   **Error Analysis:** Manual inspection reveals that most failures stem from planning limitations rather than grounding, confirming UI-Ins-7B effectively solves the grounding bottleneck in online environments.
*   **Data Scaling:** We scaled SFT data to **1M samples**. Performance improved across all benchmarks (new SOTA) while maintaining stability in the RL stage, demonstrating the robustness of our method.
*   **Systematic Instruction Analysis:** We conducted *Leave-One-Out* and *Keep-One-Left* ablations, validating the synergy and necessity of diverse instruction perspectives.
*   **RL Algorithm Comparison:** We compared GRPO, RLOO, and DAPO. Results confirm that GRPO offers the optimal trade-off between performance and training efficiency.

---

**Highlight of Contributions:**

1.  **Systematic Analysis:** We provided the first comprehensive study on instruction quality and diversity in GUI grounding, establishing the empirical foundation for the field.
2.  **Instruction-as-Reasoning Paradigm:** We proposed a novel framework that enables adaptive reasoning pathway selection for diverse scenarios.
3.  **SOTA Performance:** We achieved state-of-the-art results on 5 offline grounding benchmarks and two online benchmarks.
4.  **In-depth Analysis:**
    *   IR **resolves the challenge of effective reasoning for grounding** Given that standard reasoning often degrades grounding performance, we demonstrate how to design reasoning method that enhance rather than hinder grounding.
    *   IR **mitigates policy collapse issue** in the SFT+RL framework for End-to-End GUI grounding.
    *   IR **unlocks emergent reasoning capabilities**, allowing the model to synthesize novel perspectives.

---

**Rebuttal Updates**
* We are delighted that Reviewer zWAk recognized the novelty and effectiveness of our IR method and found our experiments helpful in the rebuttal, thus increasing the score from 2 to 6.

---

### Meta-Review · Area_Chair_r5ra · 2026-01-07

**Summary:**

reviewers gave 2,6,6,6. concerns include

Unclear choice of GRPO for RL training over other methods, lack of empirical evidence, and lack of significant novelty over standard GRPO and related RL approaches.

Lack of theory or analysis showing the optimality of the RL selection method.

Lack of analysis of LLM-generated instructions in the data synthesis pipeline. Other related concerns include lack of description of the raw dataset formats and how training is performed.

The analysis of the online evaluation experiments is relatively weak. While the overall performance improves, it is unclear which specific capabilities are enhanced; more detailed analysis is needed.

Only conducts experiments on one online agent benchmark, especially in the context of Android. This might be insufficient to prove the performance of the UI-Ins in dynamic GUI environments.

Paper writing and clarify in figures and tables can be improved.

**Reviewer Concerns:**

Unclear choice of GRPO for RL training over other methods, lack of empirical evidence, and lack of significant novelty over standard GRPO and related RL approaches.

--> authors added experiments and comparisons with standard RL approaches.

Lack of theory or analysis showing the optimality of the RL selection method.

--> not addressed, but this is a hard question.

Lack of analysis of LLM-generated instructions in the data synthesis pipeline. Other related concerns include lack of description of the raw dataset formats and how training is performed.

--> addressed with qualitative analysis of instructions and data synthesis pipeline

The analysis of the online evaluation experiments is relatively weak. While the overall performance improves, it is unclear which specific capabilities are enhanced; more detailed analysis is needed.

--> addressed

Only conducts experiments on one online agent benchmark, especially in the context of Android. This might be insufficient to prove the performance of the UI-Ins in dynamic GUI environments.

--> addressed with new experiments on OS-World

Paper writing and clarify in figures and tables can be improved.

--> addressed

**Reviewer Scores:**

zWAk seemed to be satisfied and will likely increase their score to a 4 or 6.

---

### Decision · Program_Chairs · 2026-01-26

Accept (Poster)